# Probing the Inductive Bias of Neural Networks through Learning Random Cellular Automata

**Jan Disselhoff** [* 1]   **Michael Wand** [* 1]

## Abstract

Why do neural networks generalize well on natural data? Natural data originates from processes subject to specific physical constraints, such as temporal and spatial invariance, that make it easier to learn. We investigate the sufficiency of these properties using 2D cellular automata as a controlled testbed: systems that are perfectly local, symmetric, and deterministic. We find that these conditions alone are *not sufficient* to predict the $k$-step evolution of a cellular automaton. We then examine smoothness (average sensitivity) as an additional criterion and find it predictive but still incomplete. Finally, we introduce a circuit complexity perspective, hypothesizing that natural functions are computable by small circuits. Junta coefficients, measuring the concentration of Fourier weight by interaction degree, provide a tighter predictor of learnability and a correspondence to combinatorial complexity. Across architectures (CNNs, transformers, MLPs), learnable functions are predominantly those with spectral weight concentrated at low degrees and therefore low complexity. These results would be consistent with the hypothesis that natural data is learnable because natural dynamics filters out complex, high-degree interactions.

## 1. Introduction

Why do neural networks generalize on natural data? The same basic recipe—stacked affine maps, simple nonlinearities, gradient descent from random initialization—succeeds across domains from quantum chemistry to language modeling (Goodfellow et al., 2016). This universality is puzzling.

[1] Department of Computer Science, Johannes Gutenberg University Mainz, Germany. Correspondence to: Jan Disselhoff <jadissel@uni-mainz.de>, Michael Wand <wandm@uni-mainz.de>.

*Proceedings of the $43^{rd}$ International Conference on Machine Learning*, Seoul, South Korea. PMLR 306, 2026. Copyright 2026 by the author(s).

A general function on $n$ input bits requires $2^n$ bits to specify, implying that exponential data is required to identify it without strong priors (Wolpert, 1996). The broad success of deep learning thus suggests that all of these natural phenomena share a common inductive bias, implicitly captured to some extent in neural networks. Correspondingly, understanding the nature of this bias remains a foundational question.

It is natural to ask whether physics, as the common denominator of natural phenomena, already explains the bias (Lin et al., 2017). Statistically, we could interpret (classical) physics as (random) information being molded into correlated patterns by a small set of deterministic rules (Appendix A), which are subject to constraints such as locality (local interactions) and temporal and spatial symmetry (the same laws apply everywhere and always). A universal bias might arise in different ways: Possibly, the constraints on the rules are already sufficient for learnable patterns to form reliably. Alternatively, learnable patterns may only indirectly emerge as the result of a longer-term process.

Our paper explores possible connections between properties of physics and the inductive bias of neural networks in controlled experiments with a simple model system, two-dimensional binary cellular automata (CAs): We sample CAs by setting up random rules, subjected to different sets of constraints, and measure generalization performance. If locality and symmetry sufficed for learnability, neural networks should easily learn to predict the dynamics of any CA, which has these constraints built-in. They do not; many CA rules remain unlearnable. Further classical constraints, such as special initial conditions or coarse-grained observations, do not change the picture fundamentally. We thus look for functional properties that could distinguish learnable from unlearnable CAs. Note that our scope is the short-time evolution of fixed, low-complexity rules. Long-running emergent dynamics are out of reach of this setup and we do not claim our findings extend to them.

We first examine smoothness, measured as perturbation sensitivity, and find it predictive but incomplete, leaving substantial variance in learnability unexplained. Analytically, in terms of the discrete Walsh-Fourier transform, an expansion in Boolean polynomials, sensitivity is the first moment of the power spectrum, which as a whole offers

a more comprehensive picture of combinatorial complexity: We hypothesize that learnable functions are those containing only low-degree interactions, a property that might plausibly emerge naturally from pattern formation from long-term random dynamics. We operationalize this via junta coefficients, which measure how Fourier weight is distributed across interaction degrees. This is connected to circuit complexity: functions computable by small $AC^0$ circuits must concentrate their spectral weight at low degrees by the Linial-Mansour-Nisan theorem (Linial et al., 1989).

Across architectures (CNNs, MLPs, Transformers), junta coefficients predict learnability far better than sensitivity alone, achieving $R^2 > 0.9$ at longer prediction horizons. Learnable functions consistently concentrate Fourier weight at low degrees; unlearnable functions spread weight to high degrees. Functions with low-degree spectral concentration can be expressed as low-degree Boolean polynomials, suggesting a limitation in combinatorial complexity.

Overall, we see that the combination of local and symmetric interactions with low-degree interactions (which implies low sensitivity) effectively characterizes the set of learnable patterns emerging from small-scale CAs. There have been conjectures that patterns arising spontaneously in nature on a certain scale are subject to complexity constraints (Schaper and Louis, 2014; Sharma et al., 2023; Abrahão et al., 2024), which would be consistent with finding indications of combinatorial restrictions in neural networks.

## 2. Related Work

**Universal Priors:** The paradox of universal induction has intrigued researchers for a long time, with the "no-free-lunch theorem" (Wolpert, 1996; Goldblum et al., 2024) generally resolved by an appeal to a variant of Occam's razor (Solomonoff, 1964a;b; Rissanen, 1978; MacKay, 2004; Hutter, 2005), which demands concise coding and breaks the symmetry by assuming a mathematical language. However, the emergence of simplicity from dynamics is not obvious (Wigner, 1960), but many potential connections have been proposed (Bennett, 1995), and physical models for a simplicity bias in nature (Sharma et al., 2023; Deutsch and Marletto, 2015) are still subject to ongoing debate (Abrahão et al., 2024).

**Priors of Deep Learning:** Prior knowledge can be incorporated into networks explicitly, e.g., by exploiting symmetry (Fukushima, 1969; Cohen and Welling, 2016) or multiscale modeling (e.g., pooling, (Yamaguchi et al., 1990)). But even for a basic fully connected MLP, biases are known. Notably, they learn low-frequency features more quickly (Rahaman et al., 2019). Using a tangent-linear model (NTK) of network training (Neal, 1996; Jacot et al., 2018), this bias can be understood as arising from decaying eigenvalues of

kernel-eigenfunctions (Cao et al., 2021), and is the source of various effects such as double descent (Belkin et al., 2019; Belkin, 2021), adversarial examples (Tsilivis and Kempe, 2022), or grokking (Kumar et al., 2024). The spectral bias has been extended to the Boolean hypercube via the Walsh-Fourier Transform (Yang and Salman, 2020; Gorji et al., 2023). The findings are fully congruent with ours, but do not consider dynamical systems.

**Sensitivity and Learnability:** The notion of *sensitivity* (susceptibility of a function to small input perturbations (Kahn et al., 1988)) is closely related to spectral bias (Vasudeva et al., 2025). A correlation between discrete sensitivity and generalization in neural networks has been established early on (Franco, 2006), and holds in a large empirical study for various notions of sensitivity (Novak et al., 2018). The findings have been replicated for recurrent architectures and transformers, which showed an even stronger bias (Bhattamishra et al., 2023). An NTK model provides an explanation (Vasudeva et al., 2025) by explicitly linking continuous low-frequency bias and discrete sensitivity.

**Boolean Function Analysis and Circuit Complexity:** Fourier analysis of Boolean functions (O'Donnell, 2008) provides mathematical tools to characterize function complexity. The Linial-Mansour-Nisan theorem (Linial et al., 1989) establishes that functions computable by polynomial-size $AC^0$ circuits must have Fourier weight concentrated at low degrees. Junta approximation quantifies how spectral weight is distributed across interaction degrees, connecting spectral decomposition to circuit complexity.

**Simplicity Bias:** Mingard et al. (2021) propose that the Bayesian prior of deep networks mostly coincides with the initialization distribution in function space, with good empirical matches in a Gaussian approximation. This implies an entropy bias towards commonly encoded functions, and induces a low-complexity prior (Mingard et al., 2019; 2025). In shallow networks, a direct connection to information in Boolean equations can be drawn (Mingard et al., 2026). This is consistent with our findings, but we examine connections to a dynamical model system.

**Links to Physical Dynamics:** Machine learning as a field has been influenced strongly by concepts from natural science and physics (Roberts et al., 2022; Zdeborová, 2020). In terms of causal links far-reaching hypotheses have been proposed, such as linking predicting dynamics to self-preserving intelligent structures (Friston, 2010), or dualities of learning and fundamental physics (Alexander et al., 2021). The influential article by Lin et al. (Lin et al., 2017) enumerates concrete links between physical models and structural properties of deep networks at a formal level.

**Cellular automata (CAs)** have long been used as model systems for physical dynamics, both in a concrete sense

of discretization of fundamental physics ('t Hooft, 2016) and as merely an abstract analog, as in our paper. Connections between sensitivity and complexity measures such as entropy and Lyapunov exponents have been studied by (Langton, 1990), referring to Wolfram's foundational categorization (Wolfram, 1984). CAs have also already been used as model system in studying neural networks: (Wulff and Hertz, 1992) learn single timesteps and already identified that chaotic CA rules pose significant learning challenges. (Gilpin, 2019) demonstrates theoretically and empirically that CNN architectures are capable of explicitly encoding the local rules underlying CA dynamics, asserting that given sufficient training data, CNNs can precisely replicate CA update rules. (Springer and Kenyon, 2021) show that the Turing-complete Game-of-Life is hard to learn in a setting of temporal coarse-graining. (Elser, 2021) follows up by designing a training protocol to sample good training examples. Aach et al. (2021) explored generalization across multiple CA rules, finding that CNNs could partially generalize to unseen configurations and even unseen rules within certain constraints. On the flip side, Neural Cellular Automata (Mordvintsev et al., 2020) demonstrate that strictly local iterative rule application that results in highly complex patterns can be learned. Bhamidipaty et al. (Bhamidipaty et al., 2023) use model systems for algorithm evaluation, including CAs, but their work does not aim at links to physics. Our study differs from previous work in its approach to study the connection between physically-motivated constraints and learnability.

## 3. Methods

### 3.1. Cellular Automata as a Controlled Testbed

We use two-dimensional cellular automata (CAs) as a testbed as they provide a coarse approximation of different "physics". Furthermore, they provide control over the structural properties we wish to investigate and are quick to simulate. A CA describes the evolution of a state defined over a discrete grid and time.

**Definition 3.1** (Boolean CA). Formally, let the state be a function $s : \Omega \times \mathbb{Z} \to \mathbb{B}$, where $\Omega \subset \mathbb{Z}^2$ is the spatial grid, $t \in \mathbb{Z}$ is discrete time, and $\mathbb{B} := \{0, 1\}$ represents the binary state of each cell. The evolution is governed by a local transition function $f : \mathbb{B}^k \to \mathbb{B}$ applied synchronously to all cells:

$$s(r, t + 1) = f(\mathcal{N}(r, t)) \qquad (1)$$

where the neighborhood $\mathcal{N}$ contains $k$ cells spatially adjacent to $r$ and $r$ itself at time $t$.

In our experiments, the state $s$ is defined on an $H \times W$ grid with binary values and torus topology (periodic boundary conditions). We use a $k = 9$ Moore neighborhood (the $3 \times 3$ square centered on the cell $r$). The transition function $f$ :

$\mathbb{B}^9 \to \mathbb{B}$ maps each of the $2^9 = 512$ possible neighborhood states to a next state for the central cell.

### 3.2. Learnability

We study the problem of predicting the state $s(r, t + T)$ from the full grid $s(\cdot, t)$, i.e. forecasting the automaton $T$ timesteps into the future. This task becomes rapidly intractable in the general case.

The effective input dimensionality grows quadratically with the prediction horizon. Predicting $T$ steps ahead requires considering a $(2T + 1) \times (2T + 1)$ neighborhood, yielding $2^{(2T+1)^2}$ possible input patterns. For $T = 3$, this already exceeds $2^{49} \approx 10^{14}$ configurations, far beyond what can be memorized or even observed during training.

Additionally, some 2D CAs with unlimited space and time are Turing-complete (Neumann and Burks, 1966), rendering inference tasks at unlimited time-horizons undecidable (Appendix B). Practically, this means that we cannot understand the systems behavior over long time scales. Our experiments thus all operate on short time scales, with constraints imposed on the rules that are either static in nature (e.g., locality) or manually imposed static constraints that correspond to hypothesized properties that might emerge dynamically over long time frames (low-degree interaction).

This sets the stage for the experiments: Given that brute-force memorization is impossible, learnability must arise from structure in the function itself, which, as we hypothesize, in turn would be shaped by constraints imposed on the rules. We thus sample random rule sets that meet certain constraints (either by prescribing or post-filtering), trying to find necessary and sufficient conditions for learnability.

### 3.3. Necessary Constraints: Locality and Symmetry

Before asking what makes a CA rule learnable, we observe that locality and symmetry are *necessary* conditions for any dynamics to be identifiable from finite data.

**Definition 3.2** (Identifiability). A transition function $f$ is *identifiable with $m$ samples* if $m$ input-output pairs suffice to determine $f$ uniquely.

We consider efficient identifiability with a realistically sized $m$ to be a prerequisite for learnable dynamics in a meaningful sense. At minimum, we should in principle be able to recover the rules from observations (e.g., data at $T = 1$).

**Locality.** The update rule $f$ depends only on a small, spatially contiguous neighborhood of $k$ cells. This restriction is crucial, as the space of possible transition functions grows doubly-exponentially as $2^{2^k}$ with neighborhood size (Wolfram, 1984). Without locality, the number of samples required to identify $f$ becomes astronomically large. In our

setting, $k = 9$ already yields $2^{512}$ possible rules, still vast, but reducible to 512 samples for exact identification.

**Symmetry.** The same transition rule $f$ applies identically at all spatial locations $r$ and all timesteps $t$. This spatio-temporal invariance is what permits generalization, as a single observation at one location provides information about all locations. More broadly, invariance forms the foundation of inductive reasoning by guaranteeing that experiments are reproducible and training data identically distributed. On the contrary, allowing the transition function to vary would again rapidly increase the combinatorial complexity. Each additional bit of information taken by $f$ to select between non-symmetric variants would double the identification costs.

**Complexity.** The space of possible $T$-step functions $f^{(T)}$ over a $(2T+1)^2$ receptive field is enormous. However, functions arising from iterated CA dynamics are not arbitrary. By construction, $f^{(T)}$ can be expressed as the $T$-fold composition of a simple local rule, bounding its Kolmogorov complexity by that of the base rule plus $O(\log T)$. The functions we study are therefore vastly simpler than generic Boolean functions on the same input dimension. This ensures that all candidate functions are simple enough to be learnable in principle, i.e., with respect to the hard limit set by coding length costs. Observed differences must reflect a specific inductive bias beyond raw Kolmogorov complexity.

Together, locality and symmetry reduce the identification problem from intractable to trivial for a *single* timestep. Given 512 distinct neighborhood configurations, the one-step transition function is fully determined. However, we consider the harder task of predicting $T$ steps ahead, where the effective function $f^{(T)}$ has a receptive field of $(2T+1)^2$ cells. Even with locality and symmetry, this composed function may or may not be learnable. *What additional structure separates learnable dynamics from unlearnable ones?*

### 3.4. Measuring Complexity: Perturbation Sensitivity and Spectral Structure

We measure the **Perturbation Sensitivity (PS)** of $f^{(T)}$ as a proxy for smoothness. PS quantifies how sensitive the system evolution is to small input changes:

$$\text{PS}(f^{(T)}) = \mathbb{E}_{x,\,i} \left[ \sum_j \mathbf{1}\left[ f^{(T)}(x)_j \neq f^{(T)}(x^{\oplus i})_j \right] \right] \quad (2)$$

where $x^{\oplus i}$ denotes the state $x$ with its $i$-th bit flipped. PS is efficiently computable and directly measures the average sensitivity relevant to prediction robustness (Franco, 2006).

As we use sensitivity over all time steps (not per-step), we can also interpret it as an indicator of how *chaotic* the system

is, i.e., how sensitive it is to small perturbations.

In a spectral view, sensitivity captures only the *mean* of the spectral distribution, not its shape. To characterize the full spectral structure, we use Fourier analysis of Boolean functions (O'Donnell, 2008). Any Boolean function $f : \{0,1\}^n \to \{0,1\}$ can be expressed via the Fourier-Walsh expansion:

$$f(x) = \sum_{S \subseteq [n]} \hat{f}(S)\chi_S(x) \quad (3)$$

where $\chi_S(x) = \prod_{i \in S}(2x_i - 1)$ are Walsh basis functions (Boolean monomials), and $\hat{f}(S)$ are Fourier coefficients. The *degree* of a term is $|S|$, representing the order of interaction.

We define the **junta coefficient** $W_k$ as the total Fourier weight at degree $k$ (O'Donnell, 2008):

$$W_k = \sum_{|S|=k} \hat{f}(S)^2 \quad (4)$$

By Parseval's identity, $\sum_k W_k = 1$, so $W_k$ measures the fraction of function variance explained by degree-$k$ interactions. The connection to sensitivity is now clear: PS is precisely the average spectral degree,

$$\text{PS}(f^{(T)}) = \frac{1}{n} \sum_k k \cdot W_k \quad (5)$$

Two functions with identical PS can have very different spectral profiles, which is why the full distribution of $W_k$ provides a richer characterization.

Junta coefficients have broad applicability in complexity theory. In our context, we highlight a connection to circuit complexity. The Linial-Mansour-Nisan theorem (Linial et al., 1989) establishes that functions computable by polynomial-size, constant-depth $AC^0$ circuits must have their Fourier weight concentrated at low degrees. Specifically, for circuits of size $M$ and depth $d$, most weight lies below degree $O((\log M)^d)$. Spectral concentration at low degrees thus implies low circuit complexity. If neural networks are biased toward learning simple functions, as suggested by (Mingard et al., 2025), then junta coefficients may provide a measurable proxy for learnability: functions with weight concentrated at low degrees should be easier to learn.

However, computing $W_k$ directly requires summing over $\binom{n}{k}$ subsets, which is intractable for large $n$. We instead exploit the relationship between noise sensitivity and spectral structure. The noise sensitivity at rate $\delta$ is:

$$NS_\delta(f^{(T)}) = \Pr_{x,\text{noise}}[f^{(T)}(x) \neq f^{(T)}(x \oplus \text{noise})] \quad (6)$$

where each bit is flipped independently with probability $\delta$.

Expanding in the Fourier basis:

$$NS_\delta(f^{(T)}) = \sum_{k=1}^{n} W_k \cdot (1 - (1 - 2\delta)^k) \qquad (7)$$

We measure $NS_\delta$ at geometrically-spaced $\delta$ values and use Non-Negative Least Squares (NNLS) to recover $W_k$. To handle ill-conditioning at large $k$, we aggregate weights into bins of increasing width: $W_0, W_1, W_{2\text{-}4}, W_{5\text{-}9}, W_{10\text{-}16}$, ..., ending in square numbers, yielding up to 15 bins total. Details are provided in Appendix C.

### 3.5. Network Architecture and Training

We evaluate three architecture families: CNNs, MLPs, and Transformers. All architectures share a modular design with a central block that is stacked $T$ times when predicting $T$ steps ahead. This ensures that the model capacity scales with the prediction horizon, so that any rule representable at $T = 1$ remains representable at any $T$. Blocks do *not* share weights across timesteps, allowing distinct intermediate representations at each step.

Note that increased capacity does not imply easier learning. While some rules may only become representable at larger $T$, the learning problem itself grows harder as the effective receptive field, function complexity and model size increase.

CNNs provide additional prior knowledge in comparison to the other two variants, as their local receptive fields and convolutions directly encode the locality and spatial invariance of CA dynamics (weights are *not* shared across layers). They generally perform better, permitting us to test a larger range of time steps. We therefore focus on CNNs in our main figures, but results for MLPs and Transformers are qualitatively similar and reported in Appendix D, along with detailed architecture specifications.

### 3.6. Training Data and Evaluation

Training data consists of pairs $(s_0, s_T)$ where $s_0 \in \mathbb{B}^{H \times W}$ is an initial state sampled uniformly at random, and $s_T = f^{(T)}(s_0)$ is the state after $T$ applications of the transition rule. Each training batch contains fresh random initializations, ensuring the network cannot memorize specific configurations.

The data format differs by architecture. CNNs predict the full grid $s_T$ directly from $s_0$, using circular padding to handle boundary conditions. For MLPs and Transformers, we instead create one sample per output cell: the input is the $(2T + 1) \times (2T + 1)$ neighborhood in $s_0$ that causally determines that cell, and the target is the corresponding single bit in $s_T$. This ensures all architectures face the same underlying prediction task despite their structural differences.

We evaluate performance using pixel-wise accuracy, the fraction of correctly predicted cell states on freshly sampled initial conditions. Given the vast space of possible inputs ($2^{(2T+1)^2} \approx 10^{14}$ for $T = 3$), repetition is negligible at our training budget, making this effectively equivalent to held-out generalization. To distinguish genuine learning from trivial predictions, we also report accuracy gain over baseline predictors. This is important because some CA rules converge to homogeneous states (all zeros or all ones), making high accuracy trivially achievable. We use both a majority-class baseline and a linear logistic regression baseline that predicts each cell independently from its local neighborhood in $s_0$. The offset to the linear model shows the gains due to deep learning, and captures most trivial behavior potentially induced by a random rule. The simpler majority-class baseline performs very similar, showing that most trivial behavior is due to uniform outputs. We deliberately use a linear baseline rather than stronger non-linear ones (e.g., Random Forests, XGBoost). Our goal is isolating the regime where neural-network non-linearity contributes to learning. Logistic regression captures the trivial behaviors a random CA may exhibit (near-constant outputs, majority-class prediction, linear separability in the neighborhood) in a single interpretable model, whereas a gap to a complex non-linear baseline would be hard to attribute.

## 4. Experiments

We first describe our experimental setup, including the sampling of CA rules, the range of prediction horizons, and the training procedure. We then present three sets of results that progressively refine our understanding of what predicts learnability. First the insufficiency of symmetry alone for learnability, then the partial explanatory power of perturbation sensitivity, and finally the stronger predictive performance of junta coefficients.

### 4.1. Experimental Setup

We evaluate learnability across a comprehensive set of CA rules and prediction horizons. Our primary focus is on outer-totalistic rules, where the transition function depends only on the center cell state and the sum of neighboring states. This reduces the rule space from $2^{512}$ general rules to $2^{18}$ outer-totalistic rules, of which we sample 512 uniformly at random. Outer-totalistic rules tend to have a higher density of complex automata, and are also called "Life-Like" automata, named after the famous "Game of Life" CA. We also repeat all experiments for general automata. For general rules, uniform sampling yields mostly trivial or chaotic dynamics. We instead sample uniformly over Langton's $\lambda$ parameter (Langton, 1990) for $T = 1$ and select corresponding rules, ensuring diversity across dynamical regimes.

For each rule, we train networks to predict $T \in \{2, 3, 4, 5, 6, 7\}$ timesteps ahead. CNNs and MLPs are eval-

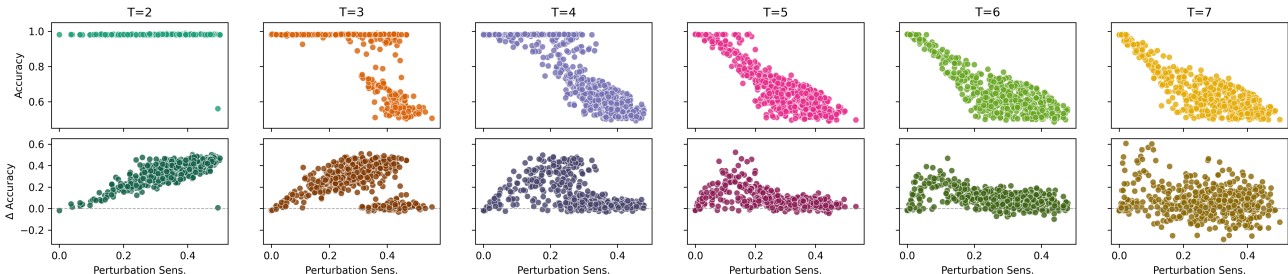

*Figure 1.* Learnability vs. perturbation sensitivity across temporal scales $T = 2$ to $T = 7$. Top row: accuracy. Bottom row: accuracy gain over majority baseline. At short horizons ($T = 2, 3$), most rules are learnable. As $T$ increases, many rules become unlearnable despite satisfying all symmetry conditions.

| | CNN | | | MLP | | | Transformer | | |
|---|---|---|---|---|---|---|---|---|---|
| T | PS | Junta | Comb. | PS | Junta | Comb. | PS | Junta | Comb. |
| 3 | $0.41 \pm 0.04$ | $0.68 \pm 0.09$ | $0.68 \pm 0.09$ | $0.71 \pm 0.05$ | $0.88 \pm 0.02$ | $0.88 \pm 0.02$ | $0.67 \pm 0.03$ | $0.81 \pm 0.02$ | $0.81 \pm 0.02$ |
| 4 | $0.71 \pm 0.03$ | $0.81 \pm 0.04$ | $0.81 \pm 0.04$ | $0.38 \pm 0.04$ | $0.80 \pm 0.02$ | $0.80 \pm 0.02$ | $0.57 \pm 0.02$ | $0.79 \pm 0.02$ | $0.79 \pm 0.02$ |
| 5 | $0.76 \pm 0.05$ | $0.89 \pm 0.03$ | $0.89 \pm 0.03$ | $0.41 \pm 0.03$ | $0.85 \pm 0.01$ | $0.85 \pm 0.01$ | $0.58 \pm 0.01$ | $0.81 \pm 0.02$ | $0.81 \pm 0.02$ |
| 6 | $0.71 \pm 0.04$ | $0.91 \pm 0.01$ | $0.91 \pm 0.01$ | $0.27 \pm 0.12$ | $0.86 \pm 0.03$ | $0.86 \pm 0.03$ | – | – | – |
| 7 | $0.70 \pm 0.04$ | $0.92 \pm 0.01$ | $0.92 \pm 0.01$ | $0.31 \pm 0.16$ | $0.86 \pm 0.02$ | $0.86 \pm 0.02$ | – | – | – |

*Table 1.* Junta coefficients substantially outperform perturbation sensitivity in predicting accuracy across architectures. The combined model offers minimal improvement over junta alone, indicating that junta coefficients subsume sensitivity information while capturing additional spectral structure. Results reported as mean $\pm$ standard deviation over 5-fold cross-validation on outer totalistic rules. $T = 6, 7$ missing for transformer models due to hardware limitations.

uated across all values of $T$, while Transformers are limited to $T \leq 5$ due to memory constraints from the quadratic attention mechanism. This yields over 3,000 training runs for our main CNN experiments alone. Each model is trained for 4096 gradient steps, using the Adam optimizer. In-depth architecture and training parameters can be found in Appendix E.

For each rule and temporal scale, we measure pixel-wise accuracy, accuracy gain over baseline, perturbation sensitivity, and junta coefficients. We focus on CNN results in the main text, as they provide the clearest architectural match to our problem setting. Results for MLPs and Transformers show qualitatively similar patterns, although with less training success, and are reported in Appendix D.

### 4.2. Symmetry Alone is Not Sufficient

We first test whether locality and spatiotemporal invariance, the symmetry properties shared by CAs and CNNs, suffice for learnability.

Figure 1 reveals that symmetry and locality are not sufficient. At short horizons ($T = 2, 3$), most rules are learnable with high accuracy. However, as $T$ increases, a substantial fraction of rules become effectively unlearnable: accuracy collapses to baseline levels despite the rules satisfying all locality and symmetry constraints. The effect is particularly pronounced at $T \geq 5$, where many perfectly local and symmetric rules remain completely unpredictable or no performance over a logistic regression baseline can be seen.

(see Table 2).

This failure is not an architectural limitation. As described in Section 3.5, network depth scales with $T$, ensuring sufficient representational capacity. The networks can represent these functions but cannot learn them from data. Something else must distinguish learnable rules from unlearnable ones.

| | CNN | | MLP | | Transformer | |
|---|---|---|---|---|---|---|
| T | Avg Acc. | $\triangle$ Acc. | Avg Acc. | $\triangle$ Acc. | Avg Acc. | $\triangle$ Acc. |
| 2 | 0.98 | 0.34 | 0.71 | 0.07 | 0.70 | 0.05 |
| 3 | 0.88 | 0.25 | 0.66 | 0.02 | 0.66 | 0.02 |
| 4 | 0.74 | 0.13 | 0.62 | 0.01 | 0.64 | 0.03 |
| 5 | 0.68 | 0.08 | 0.63 | 0.03 | 0.65 | 0.05 |
| 6 | 0.66 | 0.09 | 0.63 | 0.06 | – | – |
| 7 | 0.66 | 0.06 | 0.64 | 0.04 | – | – |

*Table 2.* Average accuracy over all rules for different models and timesteps. **Avg Acc.** shows the average accuracy over all 512 randomly sampled automata. $\triangle$ **Acc.** shows difference to average accuracy of a logistic regression baseline over the same automata. All models always outperform logistic regression on average, with CNN performing the best, but decreasing in performance with larger $T$. CNN and transformer models have difficulty outperforming logistic regression baselines.

### 4.3. Further Constraints

As symmetry and locality turn out to be insufficient constraints, we also briefly consider further restrictions:

**Coarse-grained observations:** We coarse-grain the output of the CA by replacing outputs by the majority within a

$(2k+1)\times(2k+1)$-neighborhoods for increasing $k$. Using odd neighborhoods means that we do not require tie-breaking. The network input is still done at full resolution, i.e., no input information is lost.

**Non-equilibrium, lower entropy initialization:** We initialize a rectangular region of size $(W - 2T) \times (H - 2T)$ with random i.i.d. uniformly chosen bits and leave a border of $T$ cells in state 0.

Our experiments show that these additional constraints, even with smoothness (next subsection) included, seem not to be sufficient to guarantee learnability within the short time-scales of our experimental setup (see Appendix I).

We could also consider further constraints, such as conservation laws and reversibility, but application to random CAs is challenging (see Appendix A: reversibility of a random rule is undecidable and conservation laws do not automatically emerge from symmetry for random rules). We thus move on to functional complexity parameters, which turn out to be fruitful.

### 4.4. Smoothness is Predictive but Incomplete

Figure 1 also reveals a clear relationship between perturbation sensitivity and learnability. Accuracy consistently decreases as PS increases, with the effect becoming more pronounced at larger $T$. High-sensitivity rules are systematically harder to learn, with PS providing a rough threshold for learnability at each temporal scale. We can also see that the model does not only learn to predict simple majority classes. At each timescale, low PS functions have consistent accuracy gain over a majority classification baseline.

However, sensitivity explains only part of the accuracy variance. Table 1 shows that PS alone achieves $R^2$ values of 0.40–0.76 with accuracy across temporal scales, leaving substantial variance unexplained. Rules with similar PS values can have dramatically different learnability.

This limitation reflects the fact that PS measures only the average spectral degree. Two functions with identical average degree can have very different spectral distributions: one may concentrate weight at low, learnable degrees, while another spreads weight across high degrees beyond the network's effective capacity. To capture this distinction, we turn to the full spectral structure.

### 4.5. Spectral Concentration Predicts Learnability

We extract junta coefficients $W_k$ for all training runs and fit Ridge regression models predicting accuracy from three feature sets: PS only, junta coefficients only, and both combined.

Table 1 demonstrates that junta coefficients dramatically outperform sensitivity. At $T \geq 5$, junta coefficients achieve

$R^2 = 0.89$–$0.92$, compared to $R^2 = 0.70$–$0.76$ for PS alone. A combined model provides no improvement over junta coefficients, as they subsume sensitivity while capturing additional information about spectral structure. Since PS is itself a linear combination of junta coefficients, the near-identical Combined and Junta scores are expected and serve as a sanity check for our spectral estimation: the recovered $W_k$ reproduce PS up to the precision of the fit.

However, junta coefficients are most predictive at high $T$, where our models struggle the most to distinguish themselves from the logistic regression baseline. We therefore also predict *accuracy gain over baseline* from junta coefficients. Results for CNNs are shown in Figure 2; plots for other architectures can be found in the Appendix. Even though neural networks struggle to outperform logistic regression at higher $T$, the performance gain can be predicted reasonably well from junta coefficients, confirming that spectral structure captures information about learnability specific to deep learning.

Figure 3 illustrates why junta coefficients predict learnability. Rules achieving high accuracy consistently concentrate their Fourier weight at low degrees, while rules with poor accuracy spread weight across higher degrees. This pattern holds across all temporal scales.

The Ridge regression coefficients corroborate this finding: weights for low-degree junta coefficients are positive, indicating that spectral concentration at these degrees promotes learnability, while weights for high-degree coefficients are negative. The coefficients for predicting *accuracy gain* are also interpretable: most positive weight is concentrated at low $W_k$, with the exception of $W_0$. That is, when the model achieves accuracy above baseline, it does so by capturing higher-degree interactions that logistic regression cannot.

## 5. Discussion and Conclusions

In this paper, we have studied which properties of a dynamical system make the time evolution learnable. Our approach has been mostly experimental, looking at differences among random variants of dynamical systems. We have limited ourselves to (i) a simple discrete model system in the form of small Boolean 2D cellular automata, and (ii) we have only studied their short-time behavior, thus focusing on properties that are statically ingrained in the rules and show up immediately.

We have studied which properties of a dynamical system make its time evolution learnable. Our approach has been experimental, comparing learnability across random variants of 2D Boolean cellular automata at short time horizons.

In this setting we could easily convince ourselves a priori that locality and symmetry are necessary in the sense that

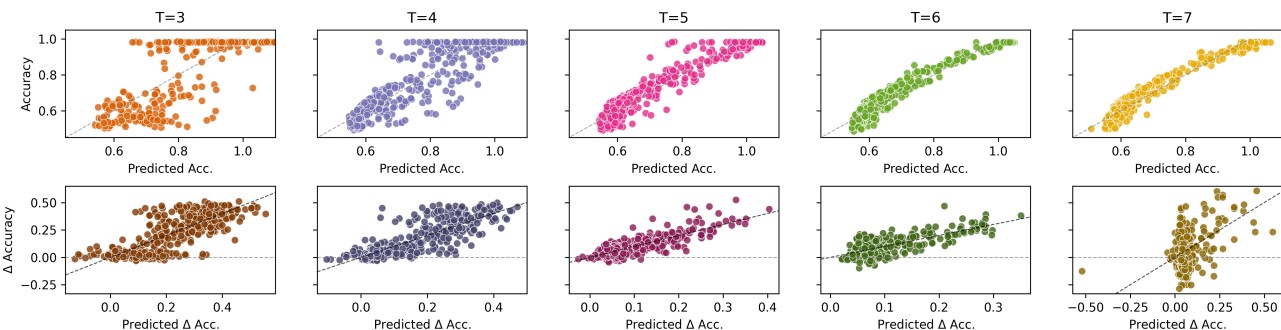

*Figure 2.* Learnability vs. predicted accuracy from $W_k$ values across temporal scales $T = 3$ to $T = 7$. Top row: accuracy. Bottom row: accuracy gain over logistic regression vs predicted accuracy gain. Dotted diagonal lines show perfect fit. All values are from held out set of 5-fold cross-validation. Junta coefficients provide a very strong predictor for accuracy, increasing in strength with growing $T$. They are also able to predict *increase* in accuracy over a logistic regression baseline, showing that they capture behavior based on the CNN architecture. Accuracy gain prediction weakens at $T = 7$, where CNN often does not learn patterns above logistic regression performance.

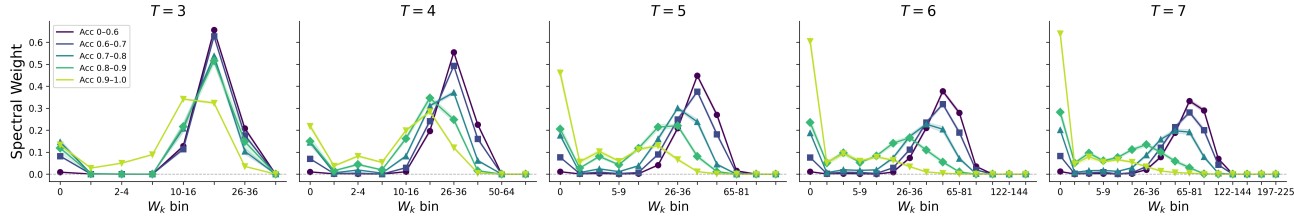

*Figure 3.* Mean spectral weight distribution for rules grouped by accuracy for outer totalistic rules and CNN architecture. Learnable rules tend to concentrate weight at low degrees, while unlearnable rules spread weight to higher degrees. We can also see that CAs tend to have very little spectral weight in low buckets except for $W_0$, making them hard to learn in general.

rules must have low-degree inputs to be even learnable in a single-time-step setup, and that symmetry is the vehicle of generalization. However, in experiments, these restrictions are not sufficient to reliably obtain learnable rules even at rather short time steps $T$. We also perform some brief checks with coarse-grained observation and lower-entropy, non-equilibrium initializations, but also could not find sufficient constraints in this way.

We then add the sensitivity of the overall time evolution to small perturbations as additional feature, i.e., varying the discrete smoothness of the function computed by the CA. This measure is strongly correlated with learnability, but low sensitivity alone is not sufficient. From a Fourier-Walsh-perspective, this is plausible, as lowering sensitivity imposes only a weak constraint on the magnitude of the higher-order coefficients, of which many exist (exponential in $n$ in medium frequencies near $k = n/2$). Thus, we might still have to deal with a intractably large set of possible functions.

The picture changes when we look at the full power spectrum $W_1, ..., W_n$ of the Fourier-Walsh series. Here, selecting a low degree subset imposes a polynomial restriction on the complexity of the functions that can be expressed. Experiments show that this constraint is actually effective: Learnability can be predicted from the power spectrum with

high accuracy, and learnability corresponds to having predominantly low-order (low $k$) interactions.

**Limitations.** Our study has significant limitations. The Boolean CA model shares only abstract structural properties with physical dynamics, and we thus cannot directly transfer findings to real-world dynamics. This also comes with the caveat that the model system might yield a very specific bias in its output data that might be coincidentally but not fundamentally correlated with the inductive bias of the network. We partially address this by measuring accuracy gains over non-deep-learning baselines, but cannot fully rule out this possibility. On the flip side, the data generator (a CA) is very similar in its working to the network (CNN/composition of random functions); seeing a related bias would thus not be unexpected. Within this study, we were not able to elucidate this further.

We are also limited to short time horizons, where static properties of the transition rule dominate. Emergent properties arising from long-time evolution may impose additional constraints on learnability that our setup does not capture. In particular, real-world structures involve complex dependencies among a large number of degrees of freedom, often involving deep scale hierarchies, which our CAs cannot generate and our networks will not try to learn; all findings reside at a fixed scale of limited input complexity. Even here,

the hypothesized connection between low-degree spectral structure and physical constraints on pattern complexity is only a motivation, providing some intuitive plausibility, but our findings obviously do not constitute rigorous evidence of a factual connection to physical dynamics.

Overall, we see our paper as an initial, small step towards trying to experimentally probe a connection between the rules of physics and the inductive bias of (deep) learning of natural patterns, which could be a promising direction for addressing some central open problems in the field.

## Impact Statement

This paper presents work whose goal is to advance the field of Machine Learning. There are many potential societal consequences of our work, none which we feel must be specifically highlighted here.

## Acknowledgements

This work has been supported by the "Research Center for Algorithmic Intelligence as an Emergent Phenomenon" funded by the Carl-Zeiss-Stiftung.

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

# Appendix

## A. Discussion: Physical Models

### A.1. Which Physical Model Should We Consider?

Our paper addresses the question of linking physical dynamics and properties of deep neural networks. As an ultimate goal, one would like to derive, experimentally and/or analytically, how to design a learning system from just the rules of physics alone, or at least recognize salient structural properties shared by deep learning methods.

Modern physics has an extremely solid experimental basis and is highly predictive for the everyday environment, which is where we would apply almost all practical deep learning methods in. In its most sophisticated form, it reduces at the fundamental level to the Standard Model (a quantum field theory incorporating special relativity) and general relativity (a non-quantum theory of curved space-time), which still seem to be at odds at extreme energy (i.e., scale) levels. While being the widely accepted foundation of all modern physics, these theories are not only mathematically involved but also difficult (general relativity) to impossible (quantum models) to access in simulations of slightly larger scale systems.

We would thus next put our focus at models from classical physics in flat space, which still explains most macroscopic mechanisms in our everyday environment very well. We should take locality, i.e., a fundamental limit of the speed at which information propagates, into account, as this could plausibly impact structure formation. Structurally speaking, we would thus consider field-theoretical models, where multi-dimensional functions over a 4D space-time are governed by partial differential equations that prescribe local (in time and space) and symmetric (frame-invariant) interactions between different such functions. Examples would include Maxwell equations or continuum formulations of fluid or solid-dynamics. If we numerically discretize such a system with a simple finite-differences approach, and represent each cell with a finite approximation of the continuous numbers involved, we would obtain a general cellular automaton (CA), which translates the original partial differential equations into local computations across spatially neighboring cells, updating subsequent temporal steps iteratively.

Our Boolean 2D CA is a further, very coarse structural abstraction, which keeps the locality and a simpler form of symmetry, but whose dynamics is far removed from realistic physics. The discrete model also cannot easily reproduce conservation laws: In the continuum case, when setting up the dynamics from a Lagrangian functional, frame-invariance automatically induces conservation of energy and momentum (Noether's theorem); while one can find analogous constructions for discrete systems (Vichniac, 1984), it is not easy to enforce specific constraints on randomly chosen rules.

One could argue that Turing-capable CAs exist, this means that the more realistic simulation of physics can be abstractly simulated even on the strongly simplified machine; however, the patterns arising do not need to bear any resemblance to the spatial patterns of the physical system.

In this sense, we have chosen a very coarse model that only shares some qualitative aspects with a "real" physical system.

### A.2. Statistical Interpretation

A classical physical system is fully specified by its rules in combination with initial conditions: All fundamental (atomistic) models of physics are time-reversible, i.e., any time-propagation operator is an invertible map on the state. Thus, we could in principle place constraints on "initial" conditions at any point in time, but for the sake of simplifying the discussion we will assume that we always start with an initialization at the beginning of the dynamical evolution considered. If we assume random initial conditions, we could characterize classical physics as a generative model that successively introduces correlations into the unstructured noise (e.g. i.i.d. for the cells in a CA) used at the beginning.

### A.3. The Role of Initial Conditions

In order for visible patterns to form in a time-reversible dynamical system, we need initial non-equilibrium conditions with limited entropy. This is easy to see for a reversible Boolean CA: If we initialize at maximum entropy, we have to maintain an encoding a specific random bit pattern drawn from a uniform distribution at every step, i.e, to store this almost certainly incompressible (Solomonoff, 1964a;b) bit pattern in each time step, which prevents the formation of geometric patterns.

In our experiments, we take this into account in two ways: When just trying to learn the $T$-step forward propagation in time, we use full-entropy random inputs as training and testing data as our focus is on learning the function, not the patterns formed. We also consider predictability under a non-equilibrium initialization (random bits restricted to a small subset of

the domain), with the obvious result that predictability grows across the boundary to empty space.

## B. Is the Prior of (our Classical Model of) Physics Decidable?

This section summarizes some basic results on decidability of inference. It formalizes concepts of learning and inference, shows that we can easily write down the physics-induced prior of our model, but that it is, in general, not decidable. All of this is obviously well-known; similar ideas have for example been discussed in the context of Solomonov induction (Solomonoff, 1964a;b; Hutter, 2005).

First, we should define the general inference task that we might be interested in:

**Definition B.1** (Inference and Priors in CAs). Let $X, Y \subset (\mathbb{Z} \times \Omega)$ subsets of the space-time grid, and $p_X, p_Y$ be probability measures on $s|_X$ and $s|_Y$, respectively. We are then tasked with the *inference problem* of computing $p_Y$ from $p_X$. If both $X$ and $Y$ are finite sets, we call it *finite inference*. $p_Y$ given $p_X \equiv$ const. is called the *prior* of the CA.

**Definition B.2** (Generative Learning and Regression in CAs). Consider sampled data $D = \{(x_i, y_i) \in (X \to \mathbb{B}) \times (Y \to \mathbb{B}) | i = 1 \ldots n\}$. Inferring an approximate probability distribution $p_Y$ from samples corresponds to *generative learning* in CAs. The special case of inferring a function $f|_{X,Y} : X \to Y$ for deterministic samples is the *regression* problem in a CA.

**Decidability** A realistic CA model of physics must be Turing capable, and as such, its behavior cannot be decided (there is no algorithm of finite size that can determine properties of its dynamic behavior (Rice, 1954)). More specifically, we can apply this to statistical learning:

**Proposition B.3** (Decidability of Inference). *The inference problem for fixed, finite $X, Y$ is decidable (by marginalization over $p_X$). The general inference problem for general CAs is undecidable, even if $X$ is finite and $Y$ is finite in space (but not time).*

**Proof sketch:**

To show this formally, we first state how we would perform inference:

**Observation B.4** (Inference). *Inference of $p_Y(s|_Y)$ in a CA can be made by marginalization over all states of $X$ that are consistent with the set $Y$ having a specified state $s|_Y$. By enumerating all of these values, we obtain $p_Y$.*

This directly leads to the decidability findings:

**For any finite and fixed input and output sets**, the algorithm above is computable: We determine the receptive field, i.e., the space-time cells that are causally connected to both X and Y (note that time might be running backwards here; we did not explicitly constrain this). This is a finite set. We can thus compute all of its possible content in finite time from $X$ by iterating over all states of $X$. Finally, we gather the desired statistics on $Y$.

This is very expensive (exponential in $|X|$), but theoretically possible.

**For outputs with infinite time:** We can ask a question of whether something will happen some time in the future, i.e., if a Turing machine (our CA) will eventually make a specific computation. This is equivalent to the problem of Rice, which is undecidable (Rice, 1954). Even answering a probabilistic question with chance of correctness of $0.5 + \epsilon, \epsilon > 0$ is undecidable (Fijalkow, 2017), which implies that the general probabilistic inference problem is undecidable, too.

**Observation B.5** (The Prior of the CA). *We can compute the prior of the CA by setting $p_X \equiv const.$ to a uniform distribution in Obs. B.4. The prior is computable for finite time horizons (i.e., for finite, fixed X,Y).*

**Discussion** The consequence of this elementary fact is that we have to distinguish between short- and long-term evolution of our dynamical system, as the latter might behave in an unforeseeable way. On a short time scale, properties inherent to the rules of the system might have an influence on learnability; this leads to a "static" regime, where we just impose rules like locality or symmetry on the simulation and observe the results in a small-scale and short-time simulation. On a long time scale, complex emergent phenomena might play a major role. This "dynamic" regime cannot be probed directly, but we postulate possible properties (sensitivity, low-order Fourier-Walsh-representation), which then constitute a heuristic approximation to the true, undecidable prior.

**Remark:** The arguments brought forward above would hold for any Turing-complete CA; it would thus for example also apply to a more realistic approximation of a classical physical field theory with Turing-complete dynamics.

## C. Estimating Junta Coefficients

We estimate junta coefficients using noise stability analysis. For each CA rule and prediction horizon $T$, we measure noise sensitivity $NS_\delta$ at 66 geometrically-spaced values of $\delta$ between 0.001 and 0.5. The geometric spacing oversamples small noise values, which provide better discrimination among high-degree coefficients where ill-conditioning is most severe.

For each $\delta$, we sample 512 random $32 \times 32$ grids and apply independent bit flips with probability $\delta$. We then evolve both the original and noised grids for $T$ timesteps and count differing output cells. With $32 \times 32 \times 512 \approx 500{,}000$ samples per $\delta$ value, the $NS_\delta$ estimates are highly stable.

We then exploit the well-established relationship between noise sensitivity and junta coefficients (O'Donnell, 2008):

$$NS_\delta(f^{(T)}) = \sum_{k=1}^{n} W_k \cdot (1 - (1 - 2\delta)^k) \tag{8}$$

This defines a linear system in the unknown $W_k$. However, the system is ill-conditioned at high $k$, since $(1 - 2\delta)^k$ values become nearly indistinguishable for large $k$ (e.g., $k = 100$ versus $k = 101$).

To address this, we bin the coefficients by degree. For consistency across different values of $T$, bin boundaries are placed at square numbers: $W_0$, $W_1$, $W_{2\text{-}4}$, $W_{5\text{-}9}$, $W_{10\text{-}16}$, and so on up to $W_{197\text{-}225}$. For each bin, we average the corresponding rows of the linear system. We also append a row enforcing the constraint $\sum_k W_k = 1$ from Parseval's identity. We chose square-aligned bin boundaries for comparability across prediction horizons $T$: since the causal receptive field at horizon $T$ contains $(2T + 1)^2$ cells, aligning bin edges to squares ensures that no bin is "half-empty" for any value of $T$.

We solve this system using SciPy's `lsq_linear` solver with bounds constraining each $W_k \in [0, 1]$. The resulting fits achieve average RMSE values between $10^{-3}$ and $4 \times 10^{-3}$, indicating good reconstruction of the noise sensitivity curve.

## D. Architecture Details

All architectures follow the modular design principle described in Section 3.5: a central block is repeated according to the prediction horizon $T$. We describe each architecture below.

### D.1. CNN

The CNN operates on full $16 \times 16$ grids, predicting all output cells simultaneously. The architecture consists of:

- An input embedding layer: a $1 \times 1$ convolution mapping the 2-channel one-hot input to 128 channels.

- $T$ sequential blocks, each containing:
    - A $3 \times 3$ convolution with circular padding (Moore neighborhood)
    - Batch normalization
    - LeakyReLU activation
    - A $1 \times 1$ convolution for additional expressivity
    - Batch normalization and LeakyReLU

- An output embedding layer: a $1 \times 1$ convolution mapping back to 2 channels (logits).

Residual connections are used, adding each block's output to its input. All intermediate layers use 128 channels.

### D.2. MLP

The MLP predicts each output cell independently from its $(2T + 1) \times (2T + 1)$ causal neighborhood. Input patches are extracted with circular padding and flattened to vectors of length $(2T + 1)^2$.

The architecture consists of:

- An input layer mapping $(2T + 1)^2$ inputs to 128 hidden units

- $3 \times T$ hidden layers, each with 128 units and ReLU activation

- An output layer mapping to 2 classes (logits)

### D.3. Transformer

The Transformer also predicts each output cell independently from its $(2T + 1) \times (2T + 1)$ neighborhood. Each cell in the patch is treated as a token.

The architecture consists of:

- A linear pixel embedding layer mapping each 2-channel cell to 128 dimensions

- A learnable CLS token prepended to the sequence

- Learnable positional embeddings for all $(2T + 1)^2 + 1$ positions

- $T$ Transformer encoder layers with 8 attention heads and embedding dimension 128

- A classification head: a linear layer mapping the CLS token output to 2 classes

The quadratic memory cost of self-attention over $(2T + 1)^2$ tokens limits the Transformer to $T \leq 5$ on our hardware.

## E. Training Details

All training runs were done on a NVIDIA GeForce RTX 4090.

All models were trained using Adam (Kingma and Ba, 2017) with default parameters except for learning rate, for 4096 update steps with binary cross-entropy loss. Hyperparameters differed slightly across architectures due to hardware and memory constraints:

| Architecture | Learning Rate | Batch Size |
|---|---|---|
| CNN | $1 \times 10^{-4}$ | 32 |
| MLP | $1 \times 10^{-4}$ | 4 |
| Transformer | $3 \times 10^{-4}$ | 1 |

Note that batch size refers to the number of $16 \times 16$ grids per update step, not the number of individual cell predictions. Since each grid contains 256 cells, the effective number of prediction targets per step is the batch size multiplied by 256. The smaller batch sizes for MLPs and Transformers reflect memory constraints from their larger per-sample computational cost.

## F. Ablation Results

Here we collect results of all model variants. In general CNNs performed the best, followed by transformer models and then MLPs, which did never strongly outperform logistic regression baseline. All timesteps contain at $512$ randomly sampled CAs.

Outer Totalistic CAs were easier to learn than arbitrary CAs, where all architectures were unable to perform much better than baseline logistic regression.

### F.1. CNN General Automata

| $T$ | PS | Junta | Combined |
|---|---|---|---|
| 3 | $0.86 \pm 0.02$ | $0.95 \pm 0.01$ | $0.95 \pm 0.01$ |
| 4 | $0.96 \pm 0.01$ | $0.98 \pm 0.00$ | $0.98 \pm 0.00$ |
| 5 | $0.98 \pm 0.00$ | $0.98 \pm 0.00$ | $0.98 \pm 0.00$ |
| 6 | $0.97 \pm 0.01$ | $0.98 \pm 0.00$ | $0.98 \pm 0.00$ |
| 7 | $0.97 \pm 0.01$ | $0.98 \pm 0.00$ | $0.98 \pm 0.00$ |

*Table 3.* 5-fold cross-validation results for predicting accuracy from PS and $W_k$ using Ridge regression. (CNN, All Automata)

| $T$ | PS | Junta | Combined |
|---|---|---|---|
| 3.0 | $0.41 \pm 0.04$ | $0.68 \pm 0.09$ | $0.68 \pm 0.09$ |
| 4.0 | $0.71 \pm 0.03$ | $0.81 \pm 0.04$ | $0.81 \pm 0.04$ |
| 5.0 | $0.76 \pm 0.05$ | $0.89 \pm 0.03$ | $0.89 \pm 0.03$ |
| 6.0 | $0.71 \pm 0.04$ | $0.91 \pm 0.01$ | $0.91 \pm 0.01$ |
| 7.0 | $0.70 \pm 0.04$ | $0.92 \pm 0.01$ | $0.92 \pm 0.01$ |

*Table 4.* 5-fold cross-validation results for predicting accuracy from PS and $W_k$ using Ridge regression. (CNN, Outer Totalistic)

## F.2. MLP Results

| $T$ | PS | Junta | Combined |
|---|---|---|---|
| 3 | $0.97 \pm 0.00$ | $0.98 \pm 0.00$ | $0.98 \pm 0.00$ |
| 4 | $0.92 \pm 0.03$ | $0.97 \pm 0.00$ | $0.97 \pm 0.00$ |
| 5 | $0.97 \pm 0.01$ | $0.98 \pm 0.00$ | $0.98 \pm 0.00$ |
| 6 | $0.90 \pm 0.08$ | $0.97 \pm 0.01$ | $0.97 \pm 0.01$ |
| 7 | $0.96 \pm 0.01$ | $0.98 \pm 0.00$ | $0.98 \pm 0.00$ |

*Table 5.* 5-fold cross-validation results for predicting accuracy from PS and $W_k$ using Ridge regression. (MLP, All Automata)

| $T$ | PS | Junta | Combined |
|---|---|---|---|
| 2 | $0.88 \pm 0.02$ | $0.90 \pm 0.02$ | $0.90 \pm 0.02$ |
| 3 | $0.71 \pm 0.05$ | $0.88 \pm 0.02$ | $0.88 \pm 0.02$ |
| 4 | $0.38 \pm 0.04$ | $0.80 \pm 0.02$ | $0.80 \pm 0.02$ |
| 5 | $0.41 \pm 0.03$ | $0.85 \pm 0.01$ | $0.85 \pm 0.01$ |
| 6 | $0.27 \pm 0.12$ | $0.86 \pm 0.03$ | $0.86 \pm 0.03$ |
| 7 | $0.31 \pm 0.16$ | $0.86 \pm 0.02$ | $0.86 \pm 0.02$ |

*Table 6.* 5-fold cross-validation results for predicting accuracy from PS and $W_k$ using Ridge regression. (MLP, Outer Totalistic)

## F.3. Transformer Results

| $T$ | PS | Junta | Combined |
|---|---|---|---|
| 2.0 | $0.91 \pm 0.02$ | $0.93 \pm 0.01$ | $0.93 \pm 0.01$ |
| 3.0 | $0.94 \pm 0.01$ | $0.95 \pm 0.01$ | $0.95 \pm 0.01$ |
| 4.0 | $0.91 \pm 0.04$ | $0.95 \pm 0.01$ | $0.95 \pm 0.01$ |
| 5.0 | $0.94 \pm 0.01$ | $0.96 \pm 0.01$ | $0.96 \pm 0.01$ |

*Table 7.* 5-fold cross-validation results for predicting accuracy from PS and $W_k$ using Ridge regression.(Transformer, All Automata)

| $T$ | PS | Junta | Combined |
|---|---|---|---|
| 2 | $0.76 \pm 0.04$ | $0.79 \pm 0.04$ | $0.79 \pm 0.04$ |
| 3 | $0.67 \pm 0.03$ | $0.81 \pm 0.02$ | $0.81 \pm 0.02$ |
| 4 | $0.57 \pm 0.02$ | $0.79 \pm 0.02$ | $0.79 \pm 0.02$ |
| 5 | $0.58 \pm 0.01$ | $0.81 \pm 0.02$ | $0.81 \pm 0.02$ |

*Table 8.* 5-fold cross-validation results for predicting accuracy from PS and $W_k$ using Ridge regression.(Transformer, Outer Totalistic)

## G. Predicting gain over logistic baseline

| | CNN | | | MLP | | | Transformer | | |
|---|---|---|---|---|---|---|---|---|---|
| T | PS | Junta | Comb. | PS | Junta | Comb. | PS | Junta | Comb. |
| 3 | $0.01 \pm 0.01$ | $0.63 \pm 0.08$ | $0.63 \pm 0.08$ | $0.03 \pm 0.05$ | $0.19 \pm 0.12$ | $0.19 \pm 0.12$ | $0.01 \pm 0.01$ | $0.11 \pm 0.03$ | $0.11 \pm 0.03$ |
| 4 | $0.17 \pm 0.07$ | $0.71 \pm 0.07$ | $0.71 \pm 0.07$ | $0.12 \pm 0.08$ | $0.51 \pm 0.13$ | $0.51 \pm 0.13$ | $-0.01 \pm 0.01$ | $0.01 \pm 0.02$ | $0.01 \pm 0.02$ |
| 5 | $0.17 \pm 0.14$ | $0.64 \pm 0.10$ | $0.64 \pm 0.10$ | $0.08 \pm 0.07$ | $0.61 \pm 0.10$ | $0.61 \pm 0.10$ | $0.00 \pm 0.02$ | $0.02 \pm 0.02$ | $0.02 \pm 0.02$ |
| 6 | $0.16 \pm 0.08$ | $0.43 \pm 0.09$ | $0.43 \pm 0.09$ | $0.01 \pm 0.03$ | $0.27 \pm 0.22$ | $0.27 \pm 0.22$ | – | – | – |
| 7 | $0.01 \pm 0.08$ | $0.11 \pm 0.20$ | $0.11 \pm 0.20$ | $-0.00 \pm 0.01$ | $0.03 \pm 0.04$ | $0.03 \pm 0.04$ | – | – | – |

Table shows accuracy of predicting accuracy gain over logistic baseline for different models, depending on if we use $PS$ only, or if we use the Junta coefficients. $PS$ is rarely able to meaningfully predict accuracy gain, while Junta coefficients can do so for CNN and MLP architectures, but fail at higher $T$.

## H. Plots for other architectures

We perform control experiments on general automata with $3 \times 3$ neighborhoods. To avoid oversampling high-PS rules, we first uniformly select a lambda parameter (ratio of patterns mapped to 1), then sample a matching rule. Plots for arbitrary CA show similar patterns to outer-totalistic experiments, with stricter PS dependence but worse overall performance. In general, the models have a much harder time improving above the logistic regression baseline. What improvement exists is still somewhat predictable from Junta coefficients. We show Figures for all architectures. Figure 4c shows relation between PS and accuracy for CNN, Figure 5c shows the same for MLPs and Figure 11c for Transformer architectures.

We also show relations between accuracy and Junta coefficients for CNN as well as predictability of accuracy gain over baseline, Figure 5d shows the same for MLPs and Figure 11d for Transformer architectures.

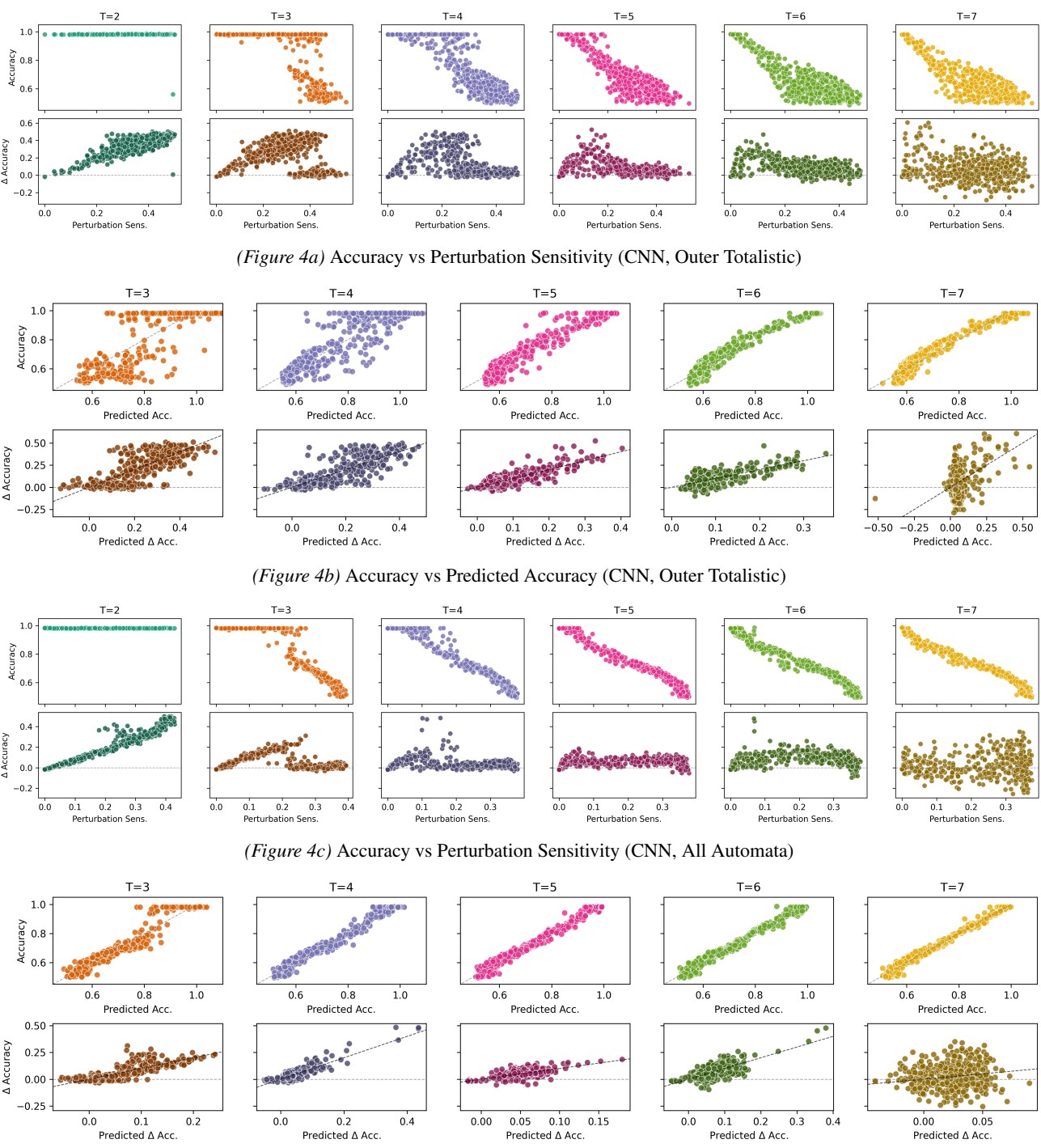

*(Figure 4a)* Accuracy vs Perturbation Sensitivity (CNN, Outer Totalistic)

*(Figure 4b)* Accuracy vs Predicted Accuracy (CNN, Outer Totalistic)

*(Figure 4c)* Accuracy vs Perturbation Sensitivity (CNN, All Automata)

*(Figure 4d)* Accuracy vs Predicted Accuracy (CNN, All Automata)

*Figure 4.* Comparison of perturbation sensitivity and junta-based predictions for outer totalistic and arbitrary automata across CNN architectures. Dotted diagonals indicate perfect prediction.

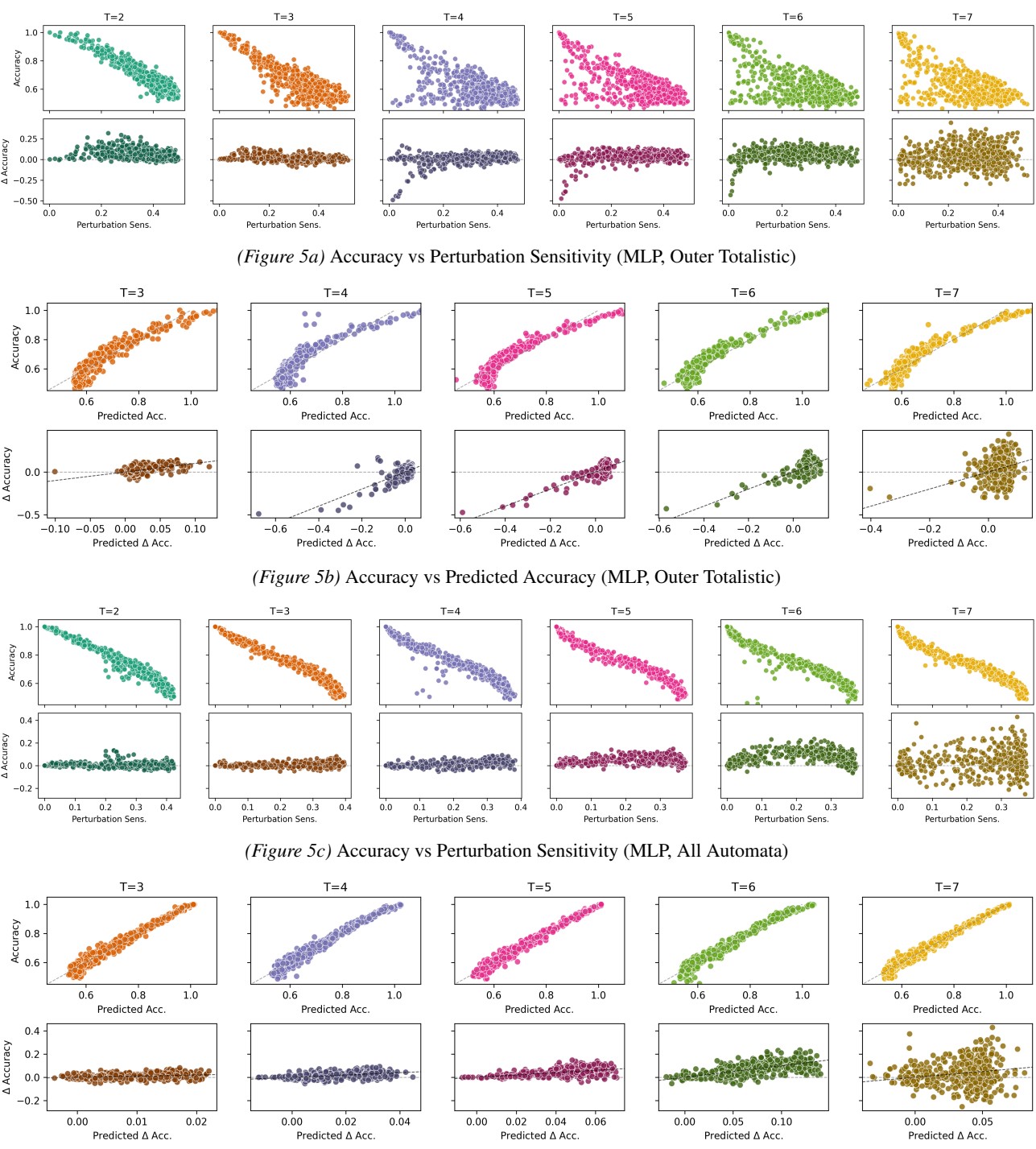

*(Figure 5a)* Accuracy vs Perturbation Sensitivity (MLP, Outer Totalistic)

*(Figure 5b)* Accuracy vs Predicted Accuracy (MLP, Outer Totalistic)

*(Figure 5c)* Accuracy vs Perturbation Sensitivity (MLP, All Automata)

*(Figure 5d)* Accuracy vs Predicted Accuracy (MLP, All Automata)

*Figure 5.* Comparison of perturbation sensitivity and junta-based predictions for outer totalistic and arbitrary automata across MLP architectures. Dotted diagonals indicate perfect prediction.

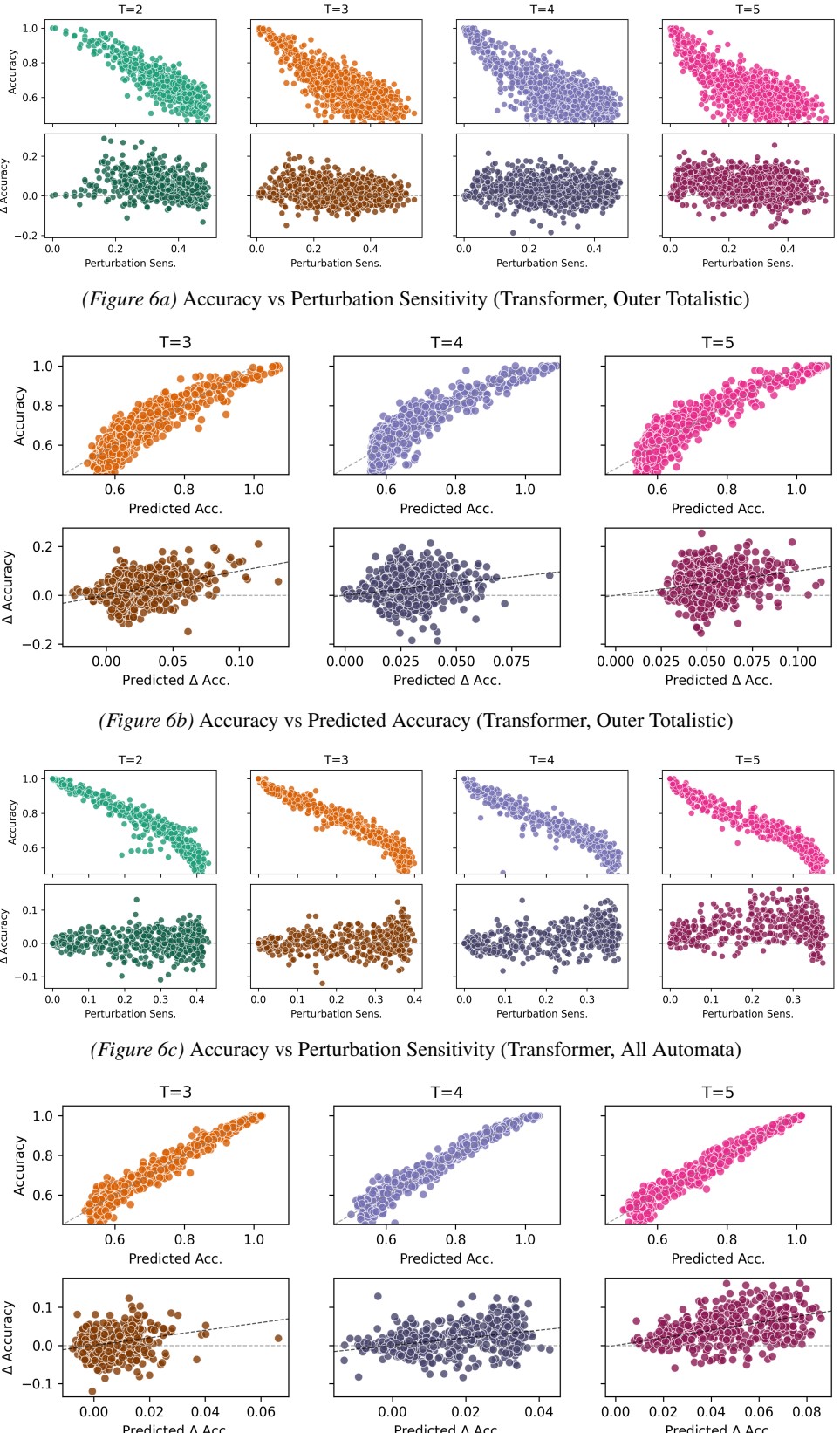

*(Figure 6a)* Accuracy vs Perturbation Sensitivity (Transformer, Outer Totalistic)

*(Figure 6b)* Accuracy vs Predicted Accuracy (Transformer, Outer Totalistic)

*(Figure 6c)* Accuracy vs Perturbation Sensitivity (Transformer, All Automata)

*(Figure 6d)* Accuracy vs Predicted Accuracy (Transformer)

*Figure 6.* Comparison of perturbation sensitivity and junta-based predictions for outer totalistic and arbitrary automata across Transformer architectures. Dotted diagonals indicate perfect prediction.

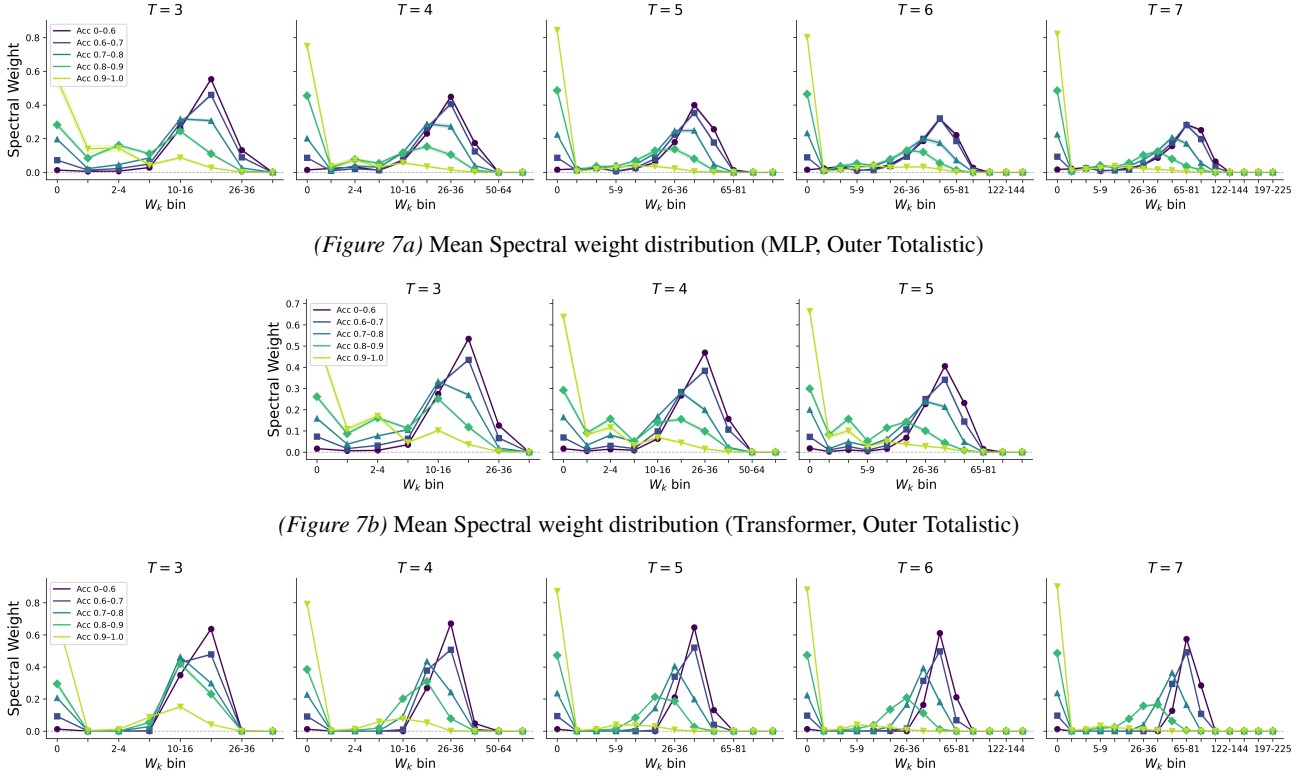

*(Figure 7a)* Mean Spectral weight distribution (MLP, Outer Totalistic)

*(Figure 7b)* Mean Spectral weight distribution (Transformer, Outer Totalistic)

*(Figure 7c)* Mean Spectral weight distribution (CNN, All Automata)

*Figure 7.* Mean spectral weight distribution by accuracy bin for MLPs and Transformers, as well as CNNs for arbitrary Automata

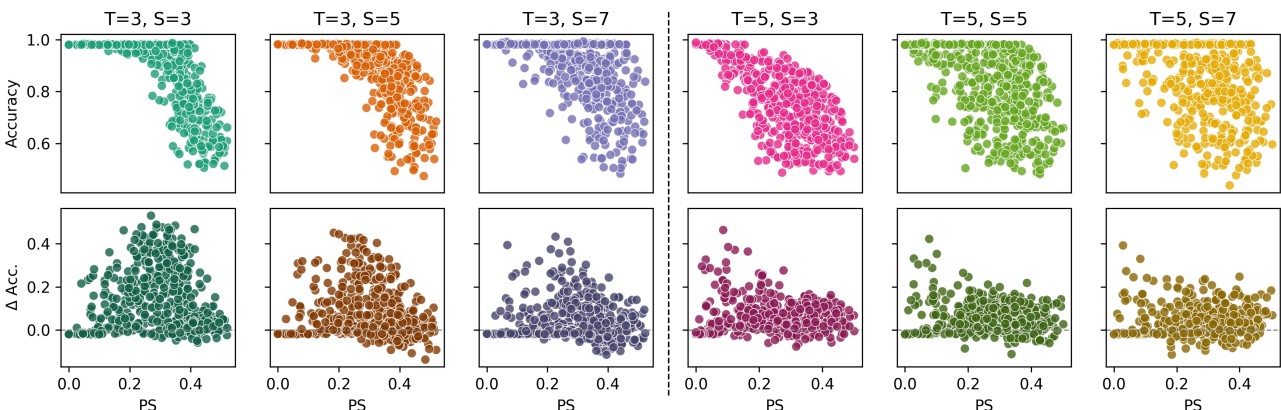

*Figure 8.* Results of different training runs for different spatial coarse grainings. From left to right, coarse graining increases from $S = 3$ to $S = 7$ with $T = 3$ left and $T = 5$ right. First row shows accuracy of the trained network, second run shows accuracy difference compared to a logistic regression baseline. While accuracy seems to increase with larger coarse graining, comparison with the baseline shows that this is mainly driven by rule simplification.

# I. Misc Plots

## I.1. Initial State

As a test of further classical restrictions to physical systems, we test whether a coarse-grained observation of a microscopic state (i.e., a discrete low-pass filter on the output of the CA) could be a discriminating feature for learnability and also whether imposing a non-equilibrium initial state with lower entropy would affect the results.

Our experiments still run at small time scales of only a few applications of the CA rules, consistent with the other experiments.

**Coarse graining** (Fig. 8) leads to less variability and thus increases average performance; in comparisons over the baseline we see that learnability still varies substantially, implying that this does not lead to sufficient constraints, nor does it reduce discriminability in terms of learnability. Note that the limitations of not probing larger scale hierarchies pointed out in the main paper apply here, too; we only study the effect of partial observation of patterns arising from a still rather short time evolution.

**Special initial conditions**, shown in Fig. 9, yield (at the small time scales considered) a transition zone where variability decreases with distance to the border (Fig. 10). The qualitative effect is not surprising, as fewer interactions with random data occur. However, we were not able to find a clear pattern of this leading to predictable learnability beyond the obvious damping of variability at the small time scales considered here.

It is generally assumed that non-equilibrium conditions are important for structure formation, and in large-scale systems (temporally and spatially), we would expect to see a strong impact, as a maximum entropy initial state would impede the formation of spatial structure, but within the time scale range of our experiments, we are not able to find significant effects beyond the obvious reduction in variability towards the border, where fewer interactions between empty space and random bits have been performed, effectively (at least) lowering $T$.

## I.2. Intermediary activations

An interesting question is *how* the network learns to represent functions. While this work is mostly concerned with the question *if* the network is able to do so, we provide some interesting analysis that might be of use in future work. In particular, we take our CNN architecture, and analyze the intermediate activations in each layer. We do so by calculating a low dimensional embedding of the activation vectors using PCA and then plotting the values.

We do so for different steps in the training process. Interestingly, the model seems to sometimes not only learn to iterated function, but seems to decompose it into the correct blocks. I.e. after the first block of the network, the PCA analysis sometimes shows a correspondence to the cellular automaton after one step. See Fig. 12.

Note that this is currently a fully visual analysis. A more rigorous test might be done by, for example, trying to predict the

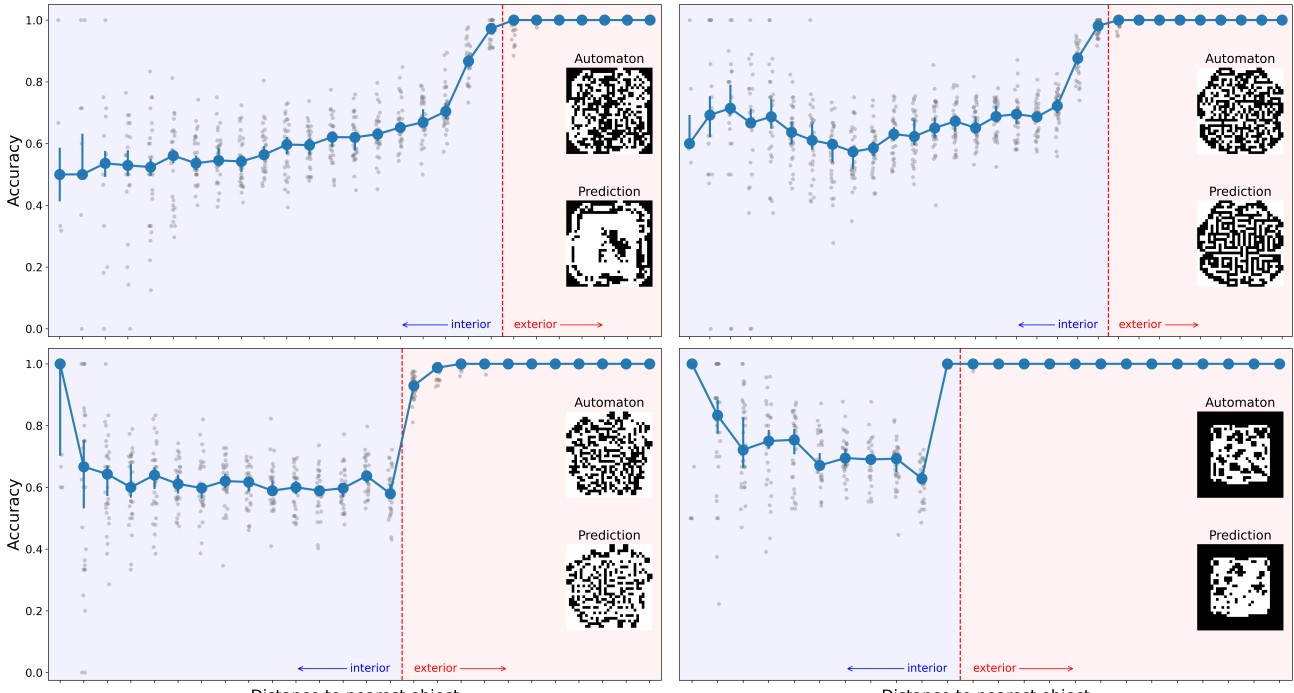

*Figure 9.* Accuracy by distance over a batch of 32 examples for 4 different automata using a CNN and localized initial conditions. Each blue dot corresponds to the median accuracy for the given distance in one such example, while the blue bars represent the surrounding 25% quartile. The light-blue regions corresponds to the interior, the red to the exterior of the object generated by the CA. On the right hand side of each example, we can see a single output of the automaton and the corresponding network prediction. While the network is able to predict the outer edges of the automaton, it is unable to infer the interior with the same accuracy.

intermediary CA states from the intermediary CNN states, without retraining the full network, and could show that the network on its own discovers simplified process dynamics that fit the full function. However, this is at the moment out of scope for this work.

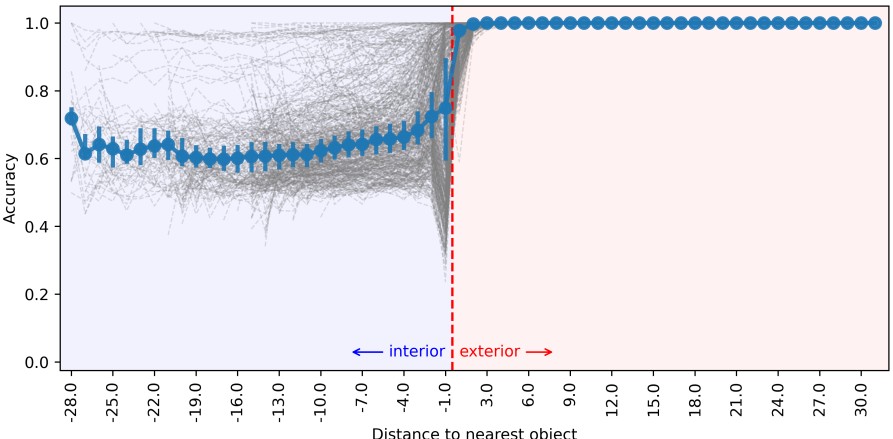

*Figure 10.* Accuracy by distance to object for 400 randomly drawn rules using a CNN and localized initial conditions. Each gray line corresponds to the mean accuracy for the given distance in one such example, calculated over a batch of 32 samples. Blue dots correspond to the median accuracy over all rules, with blue bars representing a 25% quartile around the median. The light-blue region corresponds to the interior, the light-red to the exterior. We can see that on average a CNN loses substantial amount of accuracy on the edge of the object generated by the CA, and then continues to lose accuracy further in the interior. In the innermost parts, the models regain accuracy, but this is partially an artifact of few rules even having such large negative distances.

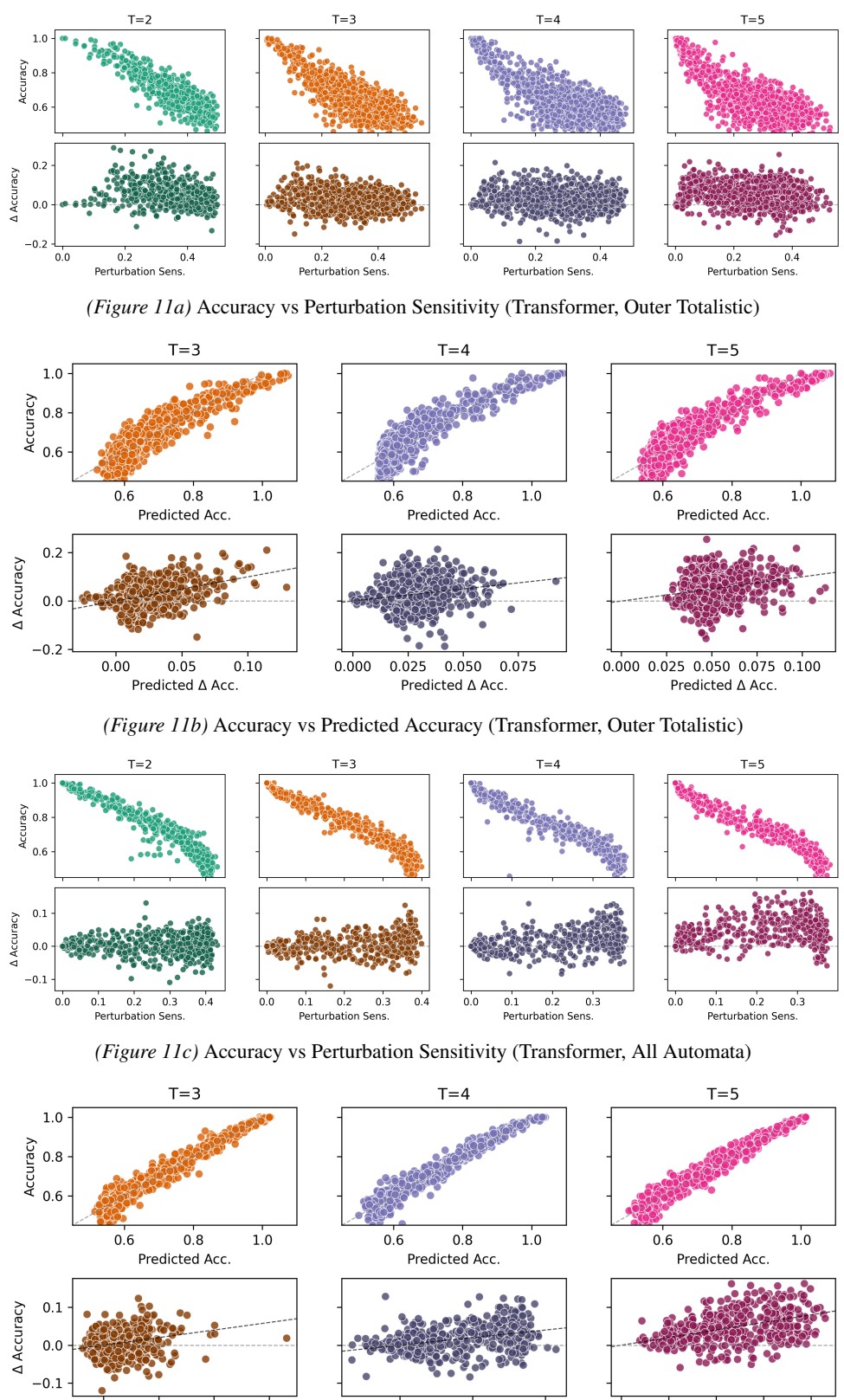

*(Figure 11a)* Accuracy vs Perturbation Sensitivity (Transformer, Outer Totalistic)

*(Figure 11b)* Accuracy vs Predicted Accuracy (Transformer, Outer Totalistic)

*(Figure 11c)* Accuracy vs Perturbation Sensitivity (Transformer, All Automata)

*(Figure 11d)* Accuracy vs Predicted Accuracy (Transformer)

*Figure 11.* Comparison of perturbation sensitivity and junta-based predictions for outer totalistic and arbitrary automata across Transformer architectures. Dotted diagonals indicate perfect prediction.

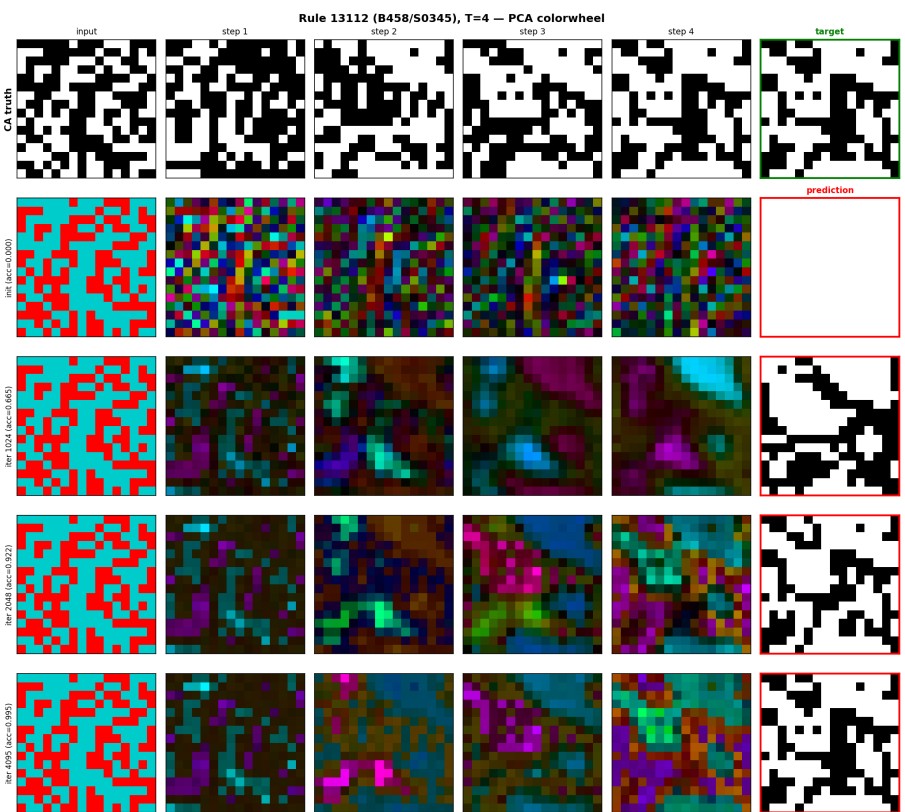

*Figure 12.* Visualization of PCA analysis of intermediary layers. First row shows an example progression of the CA, following rows correspond to time in training process at which representations were calculated (0,1024,2048 and 4096 steps). Columns show intermediary representations at Input, and then after each CNN block. Later in training, the representations seem to somewhat track the CA steps, even though intermediate steps were never part of the training.

