# OpenReview forum: "Probing the Inductive Bias of Neural Networks through Learning Random Cellular Automata"
_ICML.cc/2026/Conference — ICML 2026 regular_

### Official Review · Reviewer_8N4A · 2026-03-02

**Soundness:** 3
**Presentation:** 3
**Significance:** 2
**Originality:** 2
**Overall Recommendation:** 4
**Confidence:** 3

**Summary:**

To understand why neural networks can learn and generalize well on natural data, they study random cellular automata. They found that learnability is tightly linked to task complexity, which is defined to be junta coefficients and PS for Boolean functions. This paper mainly remains experimental.

**Compliance With Llm Reviewing Policy:**

Affirmed.

**Final Justification:**

Although I still think this paper is a bit "toy" in some sense, its analysis is well designed and may provide insights into real systems. Therefore, I raised my score from 3 to 4.

**Key Questions For Authors:**

1. What about for larger T? Right now, T = 7 is the largest time step.
2. Since PS is already a linear combination of Junta coefficients, we should expect (junta) = (junta + PS), which is indeed the case in "Figure 2" (should be "Table 2").
3. I understand that this study is experimental. Main findings are macroscopic and statistical (accuracy, junta coefficients). Can you do a more mechanistic analysis of what's happening inside the neural network? For example, explain what a convolutional filter is doing.

**Limitations:**

Yes

**Strengths And Weaknesses:**

* The paper does not fully address why neural networks can learn and generalize well on natural data, but the authors admit this limitation.
* The paper is nicely written
* This paper has experimental contributions: show that the learnability of cellular automata is linked to task complexity.
* The task and the analysis methodology are not super novel.

---

> ### Author Rebuttal · Authors · 2026-03-30
>
> We would like to thank the reviewer for their review. To address the questions:
>
> 1. Scaling to larger $T$ is a valid direction, but faces practical constraints on both sides. Computationally, training cost scales with $T$ since the input grid grows as $(2T+1)^2$. More importantly, the prediction signal itself is already diminishing. As shown in our paper, the $R^2$ for predicting CNN $\Delta$acc drops to 0.113 at $T=7$. This is not a failure of the junta coefficients per se, but reflects that at high $T$, almost all rules become too complex for the networks to learn meaningfully, leaving little variance to predict.
>
> 2. We agree with the reviewer, and we appreciate the observation. Since perturbation sensitivity is itself a linear combination of junta coefficients (weighted by degree), adding it on top of the full junta spectrum should provide little to no additional predictive power, which is exactly what Table 2 shows. We see this as supporting our argument: the junta coefficients already capture the relevant spectral information, and PS does not add anything beyond what is already contained in them.  Furthermore, the fact that the combined score is nearly identical to the junta-only score serves as a sanity check for our approximation scheme, since we can predict PS from the junta coefficients. We will make this interpretation explicit in the revised manuscript.
>
> 3. This is an interesting suggestion, but it is somewhat orthogonal to our paper's main contribution. Our goal is to characterize *which* functions neural networks can learn (via the spectral properties of the target function), not *how* they learn them internally. That said, motivated by this question, we did follow the suggestion and performed some preliminary analysis: We plotted intermediate network (CNN) activations using PCA-based dimensionality reduction and we could observe in typical individual examples that the network creates progressively more structure over the course of training. The results can be viewed at the following URLs (anonymous links):
>
>    https://ibb.co/QjYM07h4 (T=4, good accuracy, good intermediary representations)
>
>    https://ibb.co/WNMFk1PP (T=4, perfect accuracy, medium intermediary representations)
>
>    https://ibb.co/bjvsV5WL (T=5, bad accuracy, bad intermediary representations)
>
>    All figures have the same structure. The first row shows the time evolution of some CA rule.
>    The following rows show PCA based visualization of the CNN network at different times in their training, at init, 1024, 2048 and 4096 train steps. From left to right we see input embedding (which trivially reconstructs the input), and then representation after the k-th block. Finally we show the actual prediction of the network.
>
>    In some cases, it appears that the network partially follows the CA evolution internally. That is, intermediate layers seem to correspond to intermediate time steps in some cases, although this is hard to confirm definitively and requires further analysis. We would thus add these PCA plots and some light analysis as supplementary information to the appendix in the revised version. A thorough mechanistic investigation remains interesting future work, but goes beyond the scope of this paper. We thank the reviewer for the suggestion.

---

> > ### Author Rebuttal · Reviewer_8N4A · 2026-04-02
> >
> > I appreciate the authors' feedback. I've raised my score to 4.

---

### Official Review · Reviewer_GmyG · 2026-03-05

**Soundness:** 3
**Presentation:** 3
**Significance:** 3
**Originality:** 3
**Overall Recommendation:** 4
**Confidence:** 3

**Summary:**

This paper investigates which properties of a dynamical system make its time evolution learnable by neural networks. Using 2D Boolean cellular automata (CA) as a tractable model system, the authors systematically vary CA rules (both outer totalistic and arbitrary) and train three architectures (CNN, MLP, Transformer) to predict the state T steps ahead. They measure two key complexity indicators: perturbation sensitivity (a proxy for smoothness) and the Fourier Walsh spectrum (junta coefficients W_K). By comparing how well these measures predict final test accuracy, they find that the full spectral distribution—especially concentration at low degrees—dramatically outperforms sensitivity alone (R^2>0.9 for longer horizons). The work establishes that symmetry and locality are necessary but insufficient for learnability, and that a low degree spectral bias (i.e., low circuit complexity) is the essential factor enabling successful learning.

**Compliance With Llm Reviewing Policy:**

Affirmed.

**Key Questions For Authors:**

Why square-aligned bins ? The bin boundaries (1, 2–4, 5–9, …) are motivated by “ending in square numbers,” but it is not clear whether this choice is optimal or whether the results are robust to different binning strategies (e.g., logarithmic, fixed width). Could the predictive power of the junta coefficients be sensitive to the binning resolution?

	Estimation of junta coefficients from noise sensitivityRelies on solving an ill-conditioned linear system. The paper uses a soft Parseval constraint and ridge regression. How sensitive are the fitted Wk to the regularization strength(weight_penalty,ridge_alpha)andtothechoice of δ values? Could over-regularization artificially suppress high-degree coefficients and bias the results toward low-degree concentration?

	What is the role of W_0 ? In Figure 3 (and the discussion of Ridge coefficients), W0 is noted as an exception: its weight is negative when predicting accuracy gain over logistic regression. Since W0 corresponds to the constant term, a negative coefficient suggests that functions with a strong constant bias are harder to improve upon. Is this interpretation correct? Could it be that W0 is merely a proxy for the majority class baseline, and the negative weight reflects that networks cannot outperform a trivial predictor when the function is nearly constant?

	Could the results be driven by rule entropy rather than spectral structure?The authors sample rules by Langton’s λ to ensure diversity, but λ is directly related to the fraction of 1s in the truth table. How does the spectral concentration correlate with λ? Is it possible that the observed predictive power of junta coefficients is partly due to λ (or other rule statistics) that also correlate with learnability? A partial correlation analysis controlling for λ would clarify.

	Why do MLPs and Transformers perform so poorly compared to CNNs?Table 1 shows that MLPs hardly beat logistic regression, and Transformers are limited to small T. The paper attributes this to architectural inductive bias, but could it be that the training protocol (e.g., batch size, learning rate) was not optimally tuned for these architectures? For example, the Transformer uses a batch size of 1, which might hinder convergence.

	What about long-term prediction?The paper deliberately restricts to short horizons, but one of the motivations is to understand “patterns arising spontaneously in nature,” which often emerge over long times. Do the authors believe that the junta coefficients of the composed function f^{(T)}will remain predictive for large T ?  Could the spectral weight shift to higher degrees as T increases, making even initially simple rules unlearnable?

**Limitations:**

yes

**Strengths And Weaknesses:**

Strengths:

a)	The paper convincingly links the inductive bias of neural networks to the Fourier Walsh spectrum of the target function, building on prior work in Boolean function analysis and circuit complexity.

b)	Over 3000 training runs across multiple architectures and rule classes provide strong statistical evidence. The use of outer totalistic and general rules, as well as different time horizons, ensures robustness.

c)	Estimating junta coefficients via noise sensitivity and non negative least squares (with careful binning) is a clever way to handle the high dimensionality of the Boolean hypercube. The code is well structured and documented.

d)	The results show that junta coefficients subsume sensitivity information and that low degree spectral concentration is a near sufficient condition for learnability, aligning with theoretical expectations (e.g., Linial–Mansour–Nisan theorem).

e)	The work speaks to long standing questions about the origin of simplicity in nature and the “unreasonable effectiveness” of deep learning in physical domains.

Weakness:

a)	The study focuses only on short time evolution (T≤7). While this is justified by the authors’ interest in “static” rule properties, the claim that these results inform our understanding of long term emergent behavior is speculative and not tested.

b)	The Boolean CA is a very coarse discretization of physical dynamics. The authors acknowledge this in the appendix, but the leap from CA to real physical systems (e.g., PDEs) remains large; the role of conservation laws, reversibility, and continuous state spaces is not explored.

c)	Although the patterns hold across CNN, MLP, and Transformer, the MLP and Transformer results show weaker absolute performance, and for T>5 the Transformer is missing due to memory limits. The conclusions are therefore most firmly established for CNNs.

d)	The paper uses logistic regression as a baseline, but it does not control for the possibility that the junta coefficients themselves are correlated with other rule statistics (e.g., rule density, Lyapunov exponents) that might also predict learnability. A more thorough ablation or partial correlation analysis would strengthen the causal interpretation.

e)	While code is provided, the exact hyperparameters for the junta estimation (e.g., number of δ values, bin boundaries) are explained, but the sensitivity of the results to these choices is not discussed. The binning scheme (square aligned bins) is reasonable, but the paper does not justify why this particular binning was chosen over alternatives.

---

> ### Author Rebuttal · Authors · 2026-03-30
>
> We would like to thank the reviewer for their review, and would like to answer their questions:
> 1. We chose square aligned bins for comparability: Since the input size at time $T$ consists of $(2T+1)^2$ cells, the maximum Junta coefficient here is also  $(2T+1)^2$. We wanted to choose a binning that does not leave a bin "half empty" for some values of $T$, which is why we aligned it to precisely those values. We will make this more clear in the revised version
> 2. Very good question, please see the answer for Reviewer TmBT. In summary, we conducted an experiment in the case of $T=2$, where an exact calculation is tractable and show that we have a very good correlation with the estimates from noise-sensitivity.
> 3. We believe this to be a reasonable interpretation. In general it makes sense that we get a negative factor when predicting differences towards the logistic baseline, as the neural network is strictly stronger, and the $W_k$ add up to $1$. I.e. a high $W_0$ implies that there are fewer complex patterns where the network has a chance to outperform the logistic baseline.
> 4. That is an interesting question which we have immediately investigated. We have calculated Langton's λ for all rules, and redid our regression, with and without the value. We also tried predicting λ from $W_k$. Note that we used  |2λ -1|, since λ is symmetric around 0.5 for color-flips. (Otherwise correlation is trivially very low). We have three results: First, λ is not well predicted by $W_k$, with a $R^2$ of $0.27$ across rules. λ alone can be used to predict accuracy, although not well: It obtains a $R^2$ of $0.9-0.22$ depending on T. Finally, adding λ to our $W_k$ as an additionally variable shows a statistically significant increase in predictive power, although a tiny one with a $\Delta R^2$ of $0.003-0.005$ depending on $T$
> 5. Yes, this is possible, we did some hyperparameter optimization to find configurations that train quickly and well, but it is certainly possible that some combination would have worked better. The low batchsize is due to architectural constraints, but also note that the batchsize refers to the number of parallel input grids the model gets in one forward pass. Each grid however contains $256$ pixels, so even the transformer architecture trains with a signal of $256$ predictions at once.
> 6. The junta coefficients on longer timescales might be somewhat predictive, but only insofar as that they predict very simple rules. The dynamics on that timescale simply become too hard to learn effectively for our architectures. It is important to stress though (the current paper discusses this, we believe that it would be worth emphasizing more, in particular early on) that we only study dynamic systems where a fixed, low-complexity random rule runs for a small number of steps. While we cover "all" of such rules, long-running emerging phenomena could in principle acquire a structure that cannot be implemented by our set of CAs in short time spans with any possible rule. Probing such phenomena also seems to be intractable with our general setup, as the emergence of complex structures over long time periods seems to be potentially very rare and hard to trigger, not to mention trying to sample from such patterns uniformly at random. Thus our findings thus only apply to (effectively or actually) short-running processes, not to long-term evolution, which is probably far out of scope of the approach as such. We would thus be very cautious in terms of generalizing the findings to long-running processes (see also answers 1 to LS18).

---

### Official Review · Reviewer_LS18 · 2026-03-12

**Soundness:** 3
**Presentation:** 3
**Significance:** 3
**Originality:** 3
**Overall Recommendation:** 4
**Confidence:** 2

**Summary:**

This paper investigates a fundamental mystery in deep learning: the origin of neural networks' generalization power on natural data. The authors construct an idealized physical simulation environment using 2D Boolean Cellular Automata (CA) to test whether physical constraints—such as locality, symmetry, and determinism—are sufficient to ensure "learnability." The study concludes that physical constraints alone are insufficient; instead, the combinatorial complexity of the system is the deciding factor. Using Fourier analysis of Boolean functions, the authors propose that low-order interactions (measured by Junta coefficients) serve as the core metric for whether a neural network can successfully learn dynamical evolution.

**Compliance With Llm Reviewing Policy:**

Affirmed.

**Final Justification:**

After reviewing the paper alongside the rebuttal, I would give a Weak Accept.

**Key Questions For Authors:**

1. The sharp performance drop as T increases suggests NNs may be fundamentally incapable of learning dynamics with "logical depth," only "shallow correlations." How do you address this limitation?

2. If you use an architecture that deliberately violates locality (e.g., an "anti-local" convolution or a fully connected net) to learn local CA rules, how does the predictive power of the Junta coefficient change?

3. Why were more representative baselines for discrete tasks (e.g., Random Forests or symbolic search) omitted?

4. Can you provide a quantitative threshold for the Junta coefficient beyond which current mainstream architectures are guaranteed to fail?

**Limitations:**

yes

**Strengths And Weaknesses:**

## Strengths

1. **Rigorous Experimental Design:** The use of 2D CA provides a "clean room" environment that eliminates confounding variables (e.g., noise, continuous sampling bias), allowing for a focused causal analysis between physical constraints and model bias.

2. **Cross-Architecture Validation:** The experiments cover CNNs, Transformers, and MLPs, demonstrating that the preference for low-order interactions is a somewhat universal phenomenon across different inductive biases.

3. **Innovative Complexity Metric:** Introducing the Junta coefficient from Boolean function analysis provides a quantitative, theoretically grounded tool for evaluating the "learning difficulty" of dynamical systems.

4. **Discrete Extension of Spectral Bias:** The paper successfully translates the concept of "spectral bias"—traditionally studied in continuous domains (frequency)—into the realm of discrete combinatorial logic.

## Weaknesses

1. **Incremental Novelty:** The observation that neural networks prefer low-order/low-frequency functions is well-documented in continuous domains (e.g., Neural Tangent Kernel studies). The findings here feel more like a "re-translation" of existing consensus into a discrete setting rather than a groundbreaking theoretical leap.

2. **Isomorphism Bias:** The structural design of CNNs (local receptive fields, translation invariance) is essentially a parameterized approximation of CA update rules. This high degree of alignment between architecture and task leads to "idealized" results that may not reflect the model's true bias when facing complex physical systems with non-local dependencies.

3. **Performance Collapse at Scale:** Results show that for T ≥ 7, almost all models fail to significantly outperform a simple Logistic Regression baseline. This suggests the theory only holds for "shallow" evolution; once chaos or deep logical depth is involved, the proposed explanation for learnability loses its potency.

4. **Weak Baselines:** Comparing only against Logistic Regression is insufficient for a top-tier conference like ICML. The lack of comparison with strong baselines for discrete logic (e.g., symbolic regression, XGBoost, or logic synthesis tools) makes it hard to discern the unique advantages of NNs in this domain.

5. **Lack of Deep Anatomy of "Unlearnable" Rules:** While many rules were found to be unlearnable, the analysis stops at "spread-out spectral weights." The authors fail to explain *why* these rules cause gradient flow failure or representation collapse from a computational theory perspective (e.g., Wolfram classes or logical depth).

6. **Gap in Theory:** The discussion on AC⁰ circuit complexity remains heuristic. There is no rigorous mathematical proof defining the mapping between NN depth and the specific Junta coefficient threshold of a CA rule.

7. **Questionable Generalization to Reality:** There is a vast abstraction gap between Boolean CA and continuous PDEs in the real world (e.g., fluid dynamics). The lack of empirical support on noisy, continuous physical data leaves the practical utility of these conclusions in doubt.

---

> ### Author Rebuttal · Authors · 2026-03-30
>
> We would like to thank the reviewer for their review and questions. To address those:
>
> 1. We would argue this is not a limitation but rather a central finding. It demonstrates that high-logical-depth functions fall outside the inductive prior of these architectures. It is important to stress that we study short-term evolution over all possible but simple/fundamental rules. This does not exclude a deviation in behavior for systems that form structures emerging over a very long time horizon – our CAs could in principle be incapable of producing them at small T (and maybe any tractable computational effort), and thus they would not be probed. Thus, for long-running dynamics, our experiments cannot rule out that certain, very specific highly-varying rules could become learnable again. This fundamental limitation of small time-horizons is clearly stated in the submission but it might be worth stressing it in the revised paper (see also answer 6 to GmyG).
> 2. Arguably both the fully connected and transformer architectures are not local, and we see good predictability, although this is to be taken with a grain of salt due to the models also performing worse. If we were to use an "anti-local" network instead of merely non-local, we would expect it to perform even worse. However, the junta coefficients are architecture-agnostic by construction and carry no inherent bias towards locality, so there is no reason to expect that violating locality would affect their predictive power.
> 3. We chose not to use such baselines, as we want to specifically explore what types of functions a neural network can fit. We have to include a base-line to exclude trivial behavior of the random CAs, such as constant outputs or majority votes. To this end, a linear model (logistic regression) covers all of these different trivial rules but do not yet provide the combinatorial expressivity of a full non-linear network. Differences over more complex baselines would be less interpretable: E.g., it is hard to say what knowledge to extract from the fact that CNNs out- or underperform Random Forests on these tasks by x%. Note that we also are not interested in the performance of the networks on these tasks per se; they are only used as a means to an end, namely to understand in which cases neural networks perform well. In short: the linear model covers the various trivial behaviors of CAs we could think of in one model, and in practice, its gain over majority voting is very slim. We would improve the discussion in the paper to point out our motivation more clearly.
> 4. We do not believe a fixed threshold exists. We believe this to be almost certainly architecture and training time dependent. Further, a clear cutoff is not necessarily correct. When we analyze which spectral degrees are used for accuracy prediction, we see that higher degrees correlate negatively with accuracy. However, this is only a correlation and no hard cutoff.

---

> > ### Author Rebuttal · Reviewer_LS18 · 2026-04-03
> >
> > Thank you for your feedback, I decided to keep the score.

---

### Official Review · Reviewer_TmBT · 2026-03-22

**Soundness:** 3
**Presentation:** 3
**Significance:** 2
**Originality:** 4
**Overall Recommendation:** 5
**Confidence:** 4

**Summary:**

This paper asks whether fundamental structural properties in physical systems, such as locality and symmetry, are enough to make dynamical evolution learnable for neural nets. The authors design and conduct controlled experiments on randomly generated 2D cellular automata, and show how the properties are not sufficient to make the data learnable, and that perturbation sensitivity is informative yet incomplete. Then, the authors show how junta coefficients capturing the distribution of spectral weight is a better predictor, and demonstrate this claim across multiple neural net architectures.

**Compliance With Llm Reviewing Policy:**

Affirmed.

**Final Justification:**

My main concerns were satisfactorily addressed, especially the direct small-scale validation against exact Fourier spectra and the quantitative results on accuracy gain over logistic regression. So I support acceptance of this paper.

**Key Questions For Authors:**

* Can you directly compare the recovered binned junta coefficients with exact Fourier spectra in a smaller setting where exact computation is feasible?
* Can you report quantitative prediction results for accuracy gain over logistic regression, not just for raw accuracy?

**Limitations:**

Yes. The draft has a thorough and honest limitation section.

**Strengths And Weaknesses:**

Strengths:
* I like the “data-centric” approach to studying learnability, and the use of cellular automata as a controlled data-generation process is clever.
* * The initial insights that (i) locality and symmetry are not sufficient conditions for learnability, and (ii) perturbation sensitivity is an incomplete predictor of learnability, are intuitive, but I appreciate how they are demonstrated experimentally through a controlled data-generation process.
* The key takeaway, namely that junta coefficients are better predictors than perturbation sensitivity, is insightful.
* While the study is conducted primarily in a toy setup, the experiments and analysis are well designed and justify the simplicity-interpretability trade-off.

Weaknesses:
* The central spectral claim is compelling, but it relies on junta coefficients estimated indirectly from noise sensitivity, and the paper does not provide a direct validation against exact Fourier spectra in a tractable setting.
* The setup is still a toy one, and questions remain about how well the findings transfer to more realistic datasets, but the authors themselves clearly acknowledge the fact in the limitations section.
* Authors establish a strong empirical correlation between spectral concentration and learnability, but it doesn't provide a precise mechanism of why these architectures preferentially learn such rules.

---

> ### Author Rebuttal · Authors · 2026-03-30
>
> We would like to thank the reviewer for their kind review. Regarding the questions:
> 1. Comparing Junta Coefficients with exact Fourier spectra
> This is an excellent recommendation, which we immediately checked for $T=2$, where such a computation is feasible. It even revealed a small implementation bug, which caused an overestimation of $W_0$ components, but did not result in any quantitative changes in our results. Even with that bug present, our reconstruction showed strong alignment of the estimated with the exact Junta coefficients, and the alignment improved slightly after fixing the bug.
> With bug: Mean $R²$ of $0.968$, Without bug: Mean $R^2$ of $0.973$
>
> 2. Reporting quantitative results for accuracy gain over baseline
> Yes, we can add the specific numbers into the revised manuscript. We already show visual results for this in Figure 3, but we will add a table similar to Table 1, too. As can be seen in the figure, accuracy gain over baseline is easiest to predict for low T.
> For CNNs we have:
>
> | Target                     | T=3   | T=4   | T=5   | T=6   | T=7   |
> | -------------------------- | ----- | ----- | ----- | ----- | ----- |
> | **Raw accuracy (CV)**      | 0.679 | 0.807 | 0.891 | 0.915 | 0.917 |
> | **Δacc CNN−logistic (CV)** | 0.633 | 0.712 | 0.637 | 0.426 | 0.113 |
>
> For small values of $T$, the accuracy gain ist easiest to predict. For larger $T$, it becomes harder, but we still see a signal even for $T=7$.

---

> > ### Author Rebuttal · Reviewer_TmBT · 2026-04-05
> >
> > Thank you for the clear rebuttal. My main concerns were satisfactorily addressed, especially the direct small-scale validation against exact Fourier spectra and the quantitative results on accuracy gain over logistic regression. I am therefore increasing my score.

---

### Decision · Program_Chairs · 2026-04-30

**Decision:**

Accept (regular)

**Comment:**

This paper constructed an ideal physical simulation environment by using 2D cellular automata, and found the locality, symmetry and determinism are insufficient to ensure learnability. They found the Junta coefficients, a measure of low-order interactions, empirically reveal whether a network can learn dynamical evolution. Since all reviewers tend to accept this paper, I follow the reviewers’ recommendation to accept this one.